# Beyond Mahalanobis-Based Scores for Textual OOD Detection

**Pierre Colombo**
Mathématiques et Informatique pour la Complexité et les Systèmes
CentraleSupelec, Université Paris Saclay
`pierre.colombo@centralesupelec.fr`

**Eduardo D. C. Gomes**
Laboratoire des signaux et systemes
CentraleSupelec, CNRS, Université Paris-Saclay
`eduardo.dadalto@centralesupelec.fr`

**Guillaume Staerman**
Inria, CEA, Universite Paris-Saclay
`guillaume.staerman@inria.fr`

**Nathan Noiry**
althiqua.io
`noirynathan@gmail.com`

**Pablo Piantanida**
International Laboratory on Learning Systems
McGill - ETS - MILA - CNRS - CentraleSupélec, Université Paris-Saclay
`pablo.piantanida@centralesupelec.fr`

## Abstract

Deep learning methods have boosted the adoption of NLP systems in real-life applications. However, they turn out to be vulnerable to distribution shifts over time which may cause severe dysfunctions in production systems, urging practitioners to develop tools to detect out-of-distribution (OOD) samples through the lens of the neural network. In this paper, we introduce `TRUSTED`, a new OOD detector for classifiers based on Transformer architectures that meets operational requirements: it is unsupervised and fast to compute. The efficiency of `TRUSTED` relies on the fruitful idea that all hidden layers carry relevant information to detect OOD examples. Based on this, for a given input, `TRUSTED` consists in *(i)* aggregating this information and *(ii)* computing a similarity score by exploiting the training distribution, leveraging the powerful concept of *data depth*. Our extensive numerical experiments involve 51k model configurations, including various checkpoints, seeds, and datasets, and demonstrate that `TRUSTED` achieves state-of-the-art performances. In particular, it improves previous `AUROC` over 3 points.

## 1 Introduction

The number of AI systems put into production has steadily increased over the last few years. This is because advanced techniques of Machine Learning (ML) and Deep Learning (DL) have brought significant improvements over previous state-of-the-art (SOTA) methods in many areas such as finance [17, 57], transportation [59], and medicine [14, 77]. However, the increasing use of black-box models has raised concerns about their societal impact: privacy [33, 64, 74, 29], security [6, 3, 20],

36th Conference on Neural Information Processing Systems (NeurIPS 2022).

safety [41, 50, 18], fairness [5, 72], and explainability [13, 63] which became areas of active research in the ML community.

This paper is about a critical safety issue, namely Out-Of-Distribution (OOD) detection [11], which refers to a change of distribution of incoming data that may cause failures of in-production AI systems. When data are tabular and of small dimension, simple statistical methods can be efficient, and one may, for instance, monitor the mean and variance of each marginal over time. However, these traditional methods do not work anymore when data are high-dimensional and/or unstructured. Thus, the need to design new techniques that incorporate incoming data and the neural networks themselves.

Distinguishing OOD examples from in-distribution (ID) examples is challenging for modern deep neural architectures. DL models transform incoming data into latent representations from which reliable information extraction is cumbersome. Because of that, designing new tools of investigation for large pretrained models, judiciously named *foundation* models by [9], is an essential line of research for the years to come. In computer vision, methods are more mature thanks to the availability of appropriate deformation techniques that allow for sensitivity analysis. In contrast, the nature of tokens in Natural Language Processing (NLP) makes it more difficult to develop such suitable methods.

In this paper, we focus on classifiers for textual data and on the ubiquitous BERT [34], DistilBERT [83], and RoBERTa [68] architectures. Existing methods can be grouped according to their positioning with respect to the network. Some works exploit only the incoming data and compare them with in-distribution examples through likelihood ratios [42, 19]. Another line of research consists in incorporating robust constraints during training, with [48] or without [103, 60] access to some available OOD examples. Another line of research focuses on post-processing methods that can be used on any pretrained models. In our view, these are the most promising tools because typical users rely on transformers without retraining. Within post-processing methods, one can distinguish softmax-based tools that compute a confidence score based on the predicted probabilities and threshold to decide whether a sample is OOD or not. Notice that this does not require direct access to in-distribution data. The seminal work is due to Hendrycks [47] who uses the maximum soft-probability, and has been pushed further in [62, 52]. In [67], authors suggest looking one step deeper into the network, namely, to compute a confidence score based on the projections of the pre-softmax layer. More recently, [75] achieved SOTA results on transformer-based encoder by computing the Mahalanobis distance [71, 32] between a test sample and the in-distribution law [58], estimated through accessible training data points.

Nevertheless, the distance-based score is computed on the last-layer embedding only, suggesting that going deeper inside the network might improve OOD detection power. Moreover, the computation of Mahalanobis-based scores requires inverting the covariance matrix of the training data, which can be prohibitive in high dimensions. It is worth noticing that the Mahalanobis-based scores can be seen as a data depth [97, 104] through a simple re-scaling [66], that is a statistical function measuring the centrality of an observation with respect to a probability distribution. Although data depths are quite natural in the context of OOD detection, they remain overlooked by the ML community. In the present work, we rely on the recently introduced *Integrated Rank-Weighted depth* [76, 94] in order to remedy the drawbacks of the Mahalanobis-based scores for OOD detection.

## 1.1 Our contribution

We first leverage the observation introduced by previous work that *all* hidden layers of a neural network carry useful information to perform textual OOD detection. For a given input $\mathbf{x}$, our method consists in computing its average latent representation $\bar{\mathbf{x}}$ and then its OOD score through the depth score of $\bar{\mathbf{x}}$ with respect to the averaged in-distribution law (see Fig. 1 for an illustration). Notice that the ability to compute averaged latent representations crucially relies on the structure of transformers layers that share the same dimension. The depth function we are using is based on the computation of the projected ranks of the test inputs using randomly sampled directions. From a theoretical viewpoint, this novel method requires fewer assumptions on the data structure than the Mahalanobis score.

We conduct extensive numerical experiments on three transformers architectures and eight datasets and benchmark our method with previous approaches. To ensure reliable results, we introduce a new

framework for evaluating OOD detection that considers hyperparameters that were unreported before. It consists of computing performances for various choices of checkpoints and seeds, which allows us to report a variance term that makes some previous methods fall within the same performance range. Our conclusions are drawn by considering over *51k configurations*, and show that our new detector based on data depth improves SOTA methods by 3 `AUROC` points while having less variance. This result supports the intuition that OOD detection is a matter of *looking at the information available across the entire network*. Our contribution can be summarized as follows:

1. **We introduce a novel OOD detection method for textual data.** Our detector `TRUSTED`[1] relies on the full information contained in pretrained transformers and leverages the concept of data depth: a given input is detected as being in-distribution or OOD sample based on its depth score with respect to the training distribution.

2. **We conduct extensive numerical experiments** and prove that our method improves over SOTA methods. Our evaluation framework is more reliable than previous studies as it includes the variance with respect to seeds and checkpoints.

3. **We release open-source code and data to ease future research**, ensure reproducibility and reduce computation overhead.

## 2 Problem Formulation

**Training distribution and classifier.** Let us denote by $\mathcal{X}$ the textual input space. Consider the multiclass classification setup with target space $\mathcal{Y} = \{1, \ldots, C\}$ of size $C \geq 2$. We assume the dataset under consideration is made of $N \geq 1$ i.i.d. samples $(\mathbf{x}_1, y_1), \ldots, (\mathbf{x}_N, y_N)$ with probability law denoted by $p_{XY}$ and defined on $\mathcal{X} \times \mathcal{Y}$. Accordingly, we will denote by $p_X$ and $p_Y$ the marginal laws of $p_{XY}$. Finally, we denote by $f_N : \mathcal{X} \to \mathcal{Y}$ the classifier that has been trained using $(\mathbf{x}_i, y_i)$.

**Open world setting.** In real-life scenarios, the trained model $f_N$ is deployed into production and will certainly be faced with input data whose law is not $p_{XY}$. To each test point $(\mathbf{x}, y)$, we associate a variable $z \in \{0, 1\}$ such that $z = 0$ if $(\mathbf{x}, y)$ stems from $p_{XY}$ and $z = 1$ otherwise. It is worth emphasizing that in our setting $f_N$ has never been faced with OOD examples before deployment. This is usually referred to as the open-world setting. From a probabilistic viewpoint, the test set distribution of the input data $p_X^{\text{test}}$ is a mixture of in-distribution and OOD samples:

$$p_X^{\text{test}}(\mathbf{x}) = \alpha \, p_{X|Z}(\mathbf{x}|z = 1) + (1 - \alpha) \, p_{X|Z}(\mathbf{x}|z = 0),$$

where $\alpha \in (0, 1)$. In this work, we will not make any further assumptions on the proportion $\alpha$ of OOD samples and on the OOD pdf $p_{X|Z}(\mathbf{x}|z = 1)$, making the problem more difficult but at the same time more well suited for practical use. Indeed, for textual data, it does not appear to be realistic to model how a corpus can evolve.

**OOD detection.** The objective of OOD detection is to construct a similarity function $s : \mathcal{X} \to \mathbb{R}_+$ that accounts for the similarity of any element in $\mathcal{X}$ with respect to the training in-distribution. For a given test input $\mathbf{x}$, we then classify $\mathbf{x}$ as in-distribution or OOD according to the magnitude of $s(\mathbf{x})$. Therefore, one fixes a threshold $\gamma$ and classifies IN (*i.e.* $\hat{z} = 0$) if $s(\mathbf{x}) > \gamma$ or OOD (*i.e.* $\hat{z} = 1$) if $s(\mathbf{x}) \leq \gamma$. Formally, denoting $g(\cdot, \gamma)$ the decision function, we take:

$$g(\mathbf{x}, \gamma) = \left\{ \begin{array}{ll} 1 & \text{if } s\left(\mathbf{x}\right) \leq \gamma, \\ 0 & \text{if } s\left(\mathbf{x}\right) > \gamma. \end{array} \right. \tag{1}$$

**Performance evaluation.** The OOD problem is a (unbalanced) classification problem, and classically, two quantities allow to measure the performance of a method. The **false alarm rate** is the proportion of samples that are classified as OOD while they are IN. For a given threshold $\gamma$, it is theoretically given by $\Pr\left(s(\mathbf{X}) \leq \gamma \,|\, Z = 0\right)$. The **true detection rate** is the proportion of samples that are predicted OOD while being OOD. For a given threshold $\gamma$, it is theoretically given by $\Pr\left(s(\mathbf{X}) \leq \gamma \,|\, Z = 1\right)$.

There exist several ways to measure the effectiveness of an OOD method. We will focus on four metrics. The first two are specifically designed to assess the quality of the similarity function $s$.

---

[1] TRUSTED stands for de**T**ecto**R** **US**ing in**T**egrated rank-w**E**ighted **D**epth.

**Area Under the Receiver Operating Characteristic curve (AUROC) [10].** It is the area under the ROC curve $\gamma \mapsto (\Pr(s(\mathbf{X}) > \gamma \,|\, Z = 0), \Pr(s(\mathbf{X}) \leq \gamma \,|\, Z = 1))$, which plots the true detection rates against the false alarm rates. The AUROC corresponds to the probability that an in-distribution example $\mathbf{X}_{in}$ has higher score than an OOD sample $\mathbf{X}_{out}$: AUROC $= \Pr(s(\mathbf{X}_{in}) > s(\mathbf{X}_{out}))$, as can be checked from elementary computations.

**Area Under the Precision-Recall curve (AUPR-IN/AUPR-OUT) [31].** It is the area under the precision-recall curve $\gamma \mapsto (\Pr(Z = 1 \,|\, s(\mathbf{X}) \leq \gamma), \Pr(s(\mathbf{X}) \leq \gamma \,|\, Z = 1))$ which plots the recall (true detection rate) against the precision (actual proportion of OOD amongst the predicted OOD). The AUPR is more relevant to unbalanced situations.

The third metric we will use is more operational as it computes the performance at a specific threshold $\gamma$ corresponding to a security requirement.

**False Positive Rate at $95\%$ True Positive Rate (FPR).** In practice, one wishes to achieve reasonable level of OOD detection. For a desired detection rate $r$, this incites to fix a threshold $\gamma_r$ such that the corresponding TPR equals $r$. At this threshold, one then computes:

$$\Pr(s(\mathbf{X}) \geq \gamma_r \,|\, z = 0) \quad \text{with} \ \ \gamma_r \ \ \text{s.t.} \ \ \text{TPR}(\gamma_r) = r. \tag{2}$$

In our work, we set $r = 0.95$ in (2).

**Error of the best classifier (Err (%)).** This refers to the lowest classification error obtained by choosing the best threshold.

# 3 TRUSTED: Textual OOD-Detection using Integrated Rank-Weighted Depth

In this work, we focus on OOD detection when using a contextual encoder (*e.g.*, BERT). We denote by $\{\phi_1, \ldots, \phi_L\}$ the $L$ functions corresponding to the layers of the encoder: for every $1 \leq l \leq L$ and a given textual input $\mathbf{x}$, $\phi_l(\mathbf{x}) \in \mathbb{R}^d$ is the embedding of $\mathbf{x}$ in the $l$-th layer, where $d$ is the dimension of the corresponding embedding space. Notice that all layers share the same dimension $d$.

## 3.1 TRUSTED in a nutshell.

Our OOD detection method is composed of three steps. For a given input $\mathbf{x}$ with predicted label $\hat{y}$:

1. We first aggregate the latent representations of $\mathbf{x}$ via an aggregation function $F : \left(\mathbb{R}^d\right)^L \to \mathbb{R}^d$. We choose to take the mean and compute

$$F_{\text{PM}}(\mathbf{x}) := F(\phi_1(\mathbf{x}), \ldots, \phi_L(\mathbf{x})) = \frac{1}{L} \sum_{l=1}^{L} \phi_l(\mathbf{x}) := \overline{\mathbf{x}}. \tag{3}$$

We will further elaborate on this choice of aggregation function in Sec. 3.2.

2. We compute a similarity score $D(F_{\text{PM}}(\mathbf{x}), F_{\text{PM}}(\mathcal{S}_{n,\hat{y}}^{\text{train}}))$ between $F_{\text{PM}}(\mathbf{x})$ and the distribution of the mean-aggregation of the training distribution samples with same predicted target as $\mathbf{x}$ (*i.e.* $\hat{y}$) that we denote by $F_{\text{PM}}(\mathcal{S}_{n,\hat{y}}^{\text{train}})$. Formally, if $\mathbf{x}_1, \ldots, \mathbf{x}_n$ are the training data, this distribution is given by $(1/n_{\hat{y}}) \sum_{i:\hat{y}_i=\hat{y}} \delta_{F_{\text{PM}}(\mathbf{x}_i)}$, with $n_{\hat{y}} = |\{i : \hat{y}_i = \hat{y}\}|$ and $\delta_x$ is the Dirac measure in $x$. We take as similarity score $D$ a depth function, namely the integrated rank-weighted depth, that we introduce in Sec. 3.3.

3. The last step consists in thresholding the previous similarity score $D(F_{\text{PM}}(\mathbf{x}), F_{\text{PM}}(\mathcal{S}_{n,\hat{y}}^{\text{train}}))$: under a given threshold $\gamma$, we classify $\mathbf{x}$ as an OOD example.

## 3.2 Layer aggregation choice

Most recent work in textual OOD detection with a pretrained transformer solely relies on the last layer of the encoder [100, 75]. Although detectors using information available in multiple layers have been proposed previously, mostly for image data, they rely on post-score aggregation heuristics that are either supervised [58, 44] (and thus require having access to OOD samples) or heavily use arbitrary heuristics [85]. TRUSTED differentiates from previous OOD detection methods as it relies on a pre-score aggregation function.

Most popular layer aggregation techniques for Transformer based architecture involve either Power Means [46, 82] or Wasserstein barycenters [26]. Motivated by both simplicity and computational efficiency, we discard the Wasserstein barycenters and decide to work with Power Mean (case $p = 1$).

### 3.3 OOD Score Computation via Integrated Rank-Weighted Depth

Since its introduction by John Tukey in 1975 [97] to extend the notion of median to the multivariate setting, the concept of statistical depth has become increasingly popular in multivariate data analysis. Multivariate data depths are nonparametric statistics that measure the centrality of any element of $\mathbb{R}^d$, where $d \geq 2$, w.r.t. a probability distribution (respectively a random variable) defined on any subset of $\mathbb{R}^d$. Let $X$ be a random variable. We denote by $P_X$ the law of $X$. Formally, a data depth is defined as follows:

$$D: \quad \begin{aligned} \mathbb{R}^d \times \mathcal{P}(\mathbb{R}^d) &\longrightarrow [0, 1], \\ (\mathbf{x}, P_X) &\longmapsto D(\mathbf{x}, P_X). \end{aligned} \tag{4}$$

The higher $D(\mathbf{x}, P_X)$, the deeper $\mathbf{x}$ is in $P_X$. Data depth finds many applications in statistics and ML ranging from anomaly detection [87, 80, 92, 96, 91] to regression [79, 45] and text automatic evaluation [95]. Numerous definitions have been proposed, such as, among others, the halfspace depth [97], the simplicial depth [65], the projection depth [66] or the zonoid depth [56], see [90, Ch. 2] for an excellent account of data depth. The halfspace depth is the most popular depth function probably due to its attractive theoretical properties [37, 104]. However, it is defined as the solution of an optimization problem (over the unit hypersphere) of a non-differentiable quantity and is therefore not easy to compute in practice [81, 39]. Furthermore, it has been show in [73] that the approximation of the halfspace depth suffers from the curse of dimensionality involving statistical rates of order $\mathrm{O}((\log(n)/n)^{1/(d-1)})$ (see Equation (12) in [73]) where $n$ is the sample size. Recently, the Integrated Rank-Weighted (IRW) depth has been introduced in [76], replacing the infimum with an expectation (see also [16, 94]) in order to remedy this drawback. In contrast to the halfspace depth, it has been show in [94] that the approximation of the IRW depth doesn't suffer from the curse of dimensionality (see Corollary B.3 in [94]). The IRW depth of $\mathbf{x} \in \mathbb{R}^d$ w.r.t. to a probability distribution $P_X$ on $\mathbb{R}^d$ is given by:

$$D_{\mathrm{IRW}}(\mathbf{x}, P_X) = \int_{\mathbb{S}^{d-1}} \min \left\{ F_u \left( \langle \mathbf{u}, \mathbf{x} \rangle \right), 1 - F_u \left( \langle \mathbf{u}, \mathbf{x} \rangle \right) \right\} \mathrm{d}\mathbf{u},$$

where $F_u(t) = \mathrm{Pr}(\langle \mathbf{u}, \mathbf{X} \rangle \leq t)$ and $\mathbb{S}^{d-1}$ is the unit hypersphere. In practice, the expectation can be approximated by means of Monte-Carlo. Given a sample $\mathcal{S}_n = \{\mathbf{x}_1, \ldots, \mathbf{x}_n\}$, the approximation of the IRW depth is defined as:

$$\widetilde{D}_{\mathrm{IRW}}(\mathbf{x}, \mathcal{S}_n) = \frac{1}{n_{\mathrm{proj}}} \sum_{k=1}^{n_{\mathrm{proj}}} \min \left\{ \frac{1}{n} \sum_{i=1}^{n} \mathbb{I}\left\{ \langle \mathbf{u}_k, \mathbf{x}_i - \mathbf{x} \rangle \leq 0 \right\}, \frac{1}{n} \sum_{i=1}^{n} \mathbb{I}\left\{ \langle \mathbf{u}_k, \mathbf{x}_i - \mathbf{x} \rangle > 0 \right\} \right\},$$

where $\mathbf{u}_k \in \mathbb{S}^{d-1}$ and $n_{\mathrm{proj}}$ is the number of direction sampled on the sphere. The approximation version of the IRW depth can be computed in $\mathcal{O}(n_{\mathrm{proj}} n d)$ and is then linear in all of its parameters. In addition, the IRW depth has many appealing properties such as invariance to scale/translation transformations or robustness [76, 16]. Furthermore, it has been successfully applied to anomaly detection [94] making it a natural choice for OOD detection.

**Connection to Mahalanobis-based score.** Interestingly enough, the Mahalanobis distance [71] can be seen as a data depth via an appropriate rescaling as suggested in [66]. It measures the distance between an element in $\mathbb{R}^d$ and a probability distribution having finite expectation and invertible covariance matrix differing from the Euclidean perspective by taking account of correlations. Precisely, the Mahalanobis depth function $D_{\mathrm{M}}(\mathbf{x}, P_X)$ is defined as: $D_{\mathrm{M}}(\mathbf{x}, p_X) = \left(1 + (\mathbf{x} - \mathbb{E}[\mathbf{X}])^\top \Sigma^{-1} (\mathbf{x} - \mathbb{E}[\mathbf{X}])\right)^{-1}$, where $\Sigma^{-1}$ is the precision matrix of the r.v. $\mathbf{X}$. Even though interesting results relying on this notion have been highlighted in [75] for OOD detection, we experimentally observe better results with the IRW depth. Additionally, the Mahalanobis distance requires the first two moments to be finite and to compute $\Sigma^{-1}$ in high dimension, which can be ill-conditioned in low data regimes. Last, inverting $\Sigma$ requires $\mathcal{O}(d^3)$ operations or storing C matrix, which can become a burden when the number of classes grows [7].

**Application to** TRUSTED**.** The second step of TRUSTED uses an OOD score on the aggregated features. Thanks to its appealing properties, we choose to rely on the Integrated Rank-Weighted Depth $D_{\mathrm{IRW}}$

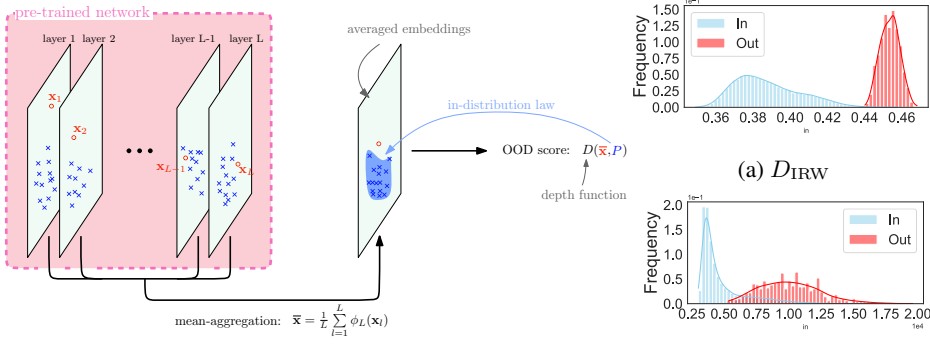

Fig. 1: TRUSTED detector. It relies on two steps: mean layer aggregation followed by the computation of $D_{\mathrm{IRW}}$.

(a) $D_{\mathrm{IRW}}$

(b) $D_{\mathrm{M}}$

Fig. 3: Histogram scores.

that measures the "similarity" between a test sample $\mathbf{x}$ and a training dataset $\mathcal{S}_n^{\mathrm{train}}$. One independent $D_{\mathrm{IRW}}$ is computed per class on the final aggregated layer. The decision is taken by taking the $D$ score of the predicted class. Formally the final score is taken as:

$$s_{\mathtt{TRUSTED}}(\mathbf{x}) = D_{\mathrm{IRW}}\left(F_{\mathrm{PM}}(\mathbf{x}), F_{\mathrm{PM}}\left(\mathcal{S}_{n,\hat{y}}^{\mathrm{train}}\right)\right), \tag{5}$$

where we recall that $F_{\mathrm{PM}}(\mathcal{S}_{n,\hat{y}}^{\mathrm{train}})$ is the distribution of the mean-aggregation of the training distribution samples with same predicted target as $\mathbf{x}$ (*i.e.* $\hat{y}$).

## 4 Experimental Settings

In this section, we first discuss the limitation of the previous works on textual OOD detection, then present the chosen benchmark, the pretrained encoders, and baseline methods.

### 4.1 Previous works and their limitations

Previous works in OOD detection [62, 84, 52, 67, 75] mostly rely on a single model to determine which methods are the best. This undermines the soundness of the conclusions that may only hold for the particular instance of the chosen model (*e.g.* for a specific checkpoint trained with a specific seed). To the best of our knowledge, no work studies the impact of the several sources of randomness that are involved, such as checkpoint and seed selections [86]). Nevertheless, these hyperparameters do impact the OOD detectors' performances. This is illustrated in Fig. 9 of the supplementary material, which gathers several Mahalanobis scores for various checkpoints of the same model.

In the light of Fig. 9, we choose to study both the impact of the checkpoint and the seed choice in our experiment. Specifically, for each model, we consider 5 different checkpoints. We save and probe models after 1k, 3k, 5k, 10k, 15k, and 20k finetuning steps. We additionally reproduce this experiment for 3 different seeds. As the reuse of checkpoints reduces the cost of research and allows for easy head-to-head comparison, our library also contains the probed models to draw general robust conclusions about the performance of the considered class of models [30, 38, 102].

### 4.2 Dataset selection

Dataset selection is instrumental for OOD detection evaluation as it is unreasonable to expect a detection method to achieve good results on any type of OOD data [1]. Since there is a lack of consensus on which benchmark to use for OOD detection in NLP, we choose to rely on the benchmark introduced by [103] which is an extension of the one proposed by [49].

**Benchmark description.** The considered benchmark is composed of three different types of in distribution datasets (referred to as IN-DS) which are used to train the classifiers: sentiment analysis (*i.e.*, SST2 [88] and IMDB [70]), topic classification (*i.e.*, 20Newsgroup [54]) and question answering (*i.e.*, TREC-10 [61]). For splitting we use either the standard split or the one provided by [103]. For the OOD datasets (referred to as OUT-DS), we first consider the aforementioned datasets (*i.e.*, any pair

of datasets can be considered as OOD). Then, we also rely on four other datasets: a concatenation of premises and respective hypotheses from two NLI datasets (*i.e.*, RTE [12, 51] and MNLI [99]), Multi30K [40] and the source of the English-German WMT16 [8]. We gather in Tab. 5 the statistics of the various data-sets and refer the reader to reference [103] for further details.

### 4.3 Baseline methods and pretrained models

**Baseline methods.** We consider the three following baselines[2]:

1. *Maximum Soft-max Probability* (MSP). This method has been proposed by [47]. Given an input $\mathbf{x}$, it relies on the final score $s_{\mathrm{MSP}}$ defined by $s_{\mathrm{MSP}}(\mathbf{x}) = 1 - \max_{y \in \mathcal{Y}} p_{Y|X}(y|\mathbf{x})$, where $p_{Y|X}(\cdot|x))$ is the soft-probability predicted by the classifier after $\mathbf{x}$ has been observed.

2. *Energy-based score (E)* [67] is defined as the score $s_E(\mathbf{x}) = T \times \log \left[ \sum_{y \in \mathcal{Y}} \exp \left( \frac{g_y(\mathbf{x})}{T} \right) \right]$, where $g_y(\mathbf{x})$ represents the logit corresponding to the class label $y$.

3. *Mahalanobis ($D_{\mathrm{M}}$)*. Following [75, 60, 103], the last layer of the encoder is considered leading to the score: $s_{\mathrm{M}}(\mathbf{x}) = -D_{\mathrm{M}}(F_{\mathrm{PM}}(\mathbf{x}), F_{\mathrm{PM}}(\mathcal{S}_{n,\hat{y}}^{\mathrm{train}}))$ where $\hat{y}$ represents the label predicted by the classifier based on the observation of $\mathbf{x}$.[3]

**Aggregation procedures.** Both $D_{\mathrm{M}}$ and $D_{\mathrm{IRW}}$ rely on feature representations of the data which are extracted from the neural networks. Our goal is to demonstrate that our aggregation procedure $F_{PM}$ defined in Eq. 3 is a relevant choice to be plugged in Eq. 5. To do so, we also perform experiments on other natural aggregation strategies we introduce in the following.

1. *Logits layer selection*. We use the raw non-normalized predictions of the classifier. In this case $F_{Logits} \equiv F(\phi_1(\mathbf{x}), \ldots, \phi_L(\mathbf{x})) = \phi_{L+1}(\mathbf{x})$.

2. *Last layer selection*. Following previous work in textual OOD detection [100], we also consider the last layer of the network. Formally $F_L \equiv F(\phi_1(\mathbf{x}), \ldots, \phi_L(\mathbf{x})) = \phi_L(\mathbf{x})$.

3. *Layer concatenation*. We follow the BERT pooler and explore the concatenation of all layers. Formally, $F_{cat} \equiv F(\phi_1(\mathbf{x}), \ldots, \phi_L(\mathbf{x})) = [\phi_1(\mathbf{x}), \cdots, \phi_L(\mathbf{x})]$ represents the concatenated vector. The main limitation of layer concatenation is that the dimension of the considered features linearly increases with the number of layers which can be problematic for very deep networks [98].

**Pretrained encoders.** To provide an exhaustive comparison, we choose to work with different types of pretrained encoders. We test the various methods on DISTILBERT (DIS.) [83], BERT [35] and ROBERTA (ROB.) [68]. We trained all models with a dropout rate [89] of 0.2, a batch size of 32, we use ADAMW [55]. Additionally, the weight decay is set to 0.01, the warmup ratio is set to 0.06 and the learning rate to $10^{-5}$.

| IN-DS | BERT Acc | DIS. Acc | ROB. Acc |
|-------|----------|----------|----------|
| 20ng  | 92.9     | 92.0     | 92.7     |
| imdb  | 91.7     | 90.6     | 93.6     |
| sst2  | 92.7     | 91.7     | 95.2     |
| trec  | 96.8     | 97.0     | 97.0     |

Tab. 1: Average test accuracy achieved by different classifier when training is initialized with different seeds.

## 5 Static Experimental Results

In this section, we demonstrate the effectiveness of our proposed detector using various pretrained models. Due to space limitations, additional tables are reported in the Supplementary Material.

### 5.1 Methods comparison

In Tab. 2, we report the aggregated score obtained by each method combined with a different aggregation function. We observe that TRUSTED obtains the best overall scores followed by $D_{\mathrm{IRW}}$ using $F_{cat}$. Similarly to previous works [75], we notice in general that score leveraging information

---

[2]Contrarily to [75], we do not use likelihood ratio as it would require using extra language models, which are not available in our setting.

[3]An alternative is to compute the minimum of all Mahalanobis distance computed on all the classes. However, we observe slightly better performance when using the predicted label $\hat{y}$.

Tab. 2: Average OOD detection performance (in %). The averages are taken over 1440 configurations and include four different `IN-DS` (20ng, imdb, sst2, trec), eight `OOD-DS`, three different seeds, five different checkpoints and three different pretrained encoders. Due to space constraints, different aggregations and related discussions are relegated to Appendix B.

| Score | Aggregation | AUROC | AUPR-IN | AUPR-OUT | FPR | Err |
|---|---|---|---|---|---|---|
| $E$ | $F_{L+1}$ | 89.9 ±9.7 | 84.9 ±19.0 | 79.9 ±27.3 | 44.9 ±33.4 | 23.9 ±22.0 |
| MSP | Soft. | 89.7 ±9.1 | 84.3 ±18.9 | 80.4 ±25.6 | 45.5 ±29.5 | 25.4 ±21.4 |
| $D_M$ | $F_L$ | 93.8 ±9.8 | 89.2 ±20.1 | 91.5 ±16.4 | 19.8 ±23.7 | 12.7 ±17.0 |
|  | $F_{L+1}$ | 71.7 ±13.7 | 54.7 ±32.0 | 73.3 ±28.4 | 62.6 ±23.1 | 37.0 ±22.9 |
|  | $F_{[L,L+1]}$ | 81.7 ±20.7 | 60.7 ±20.0 | 83.8 ±20.3 | 73.4 ±23.5 | 33.0 ±21.3 |
|  | $F_{L \oplus L+1}$ | 83.6 ±10.6 | 61.9 ±39.3 | 79.4 ±26.9 | 81.5 ±10.1 | 30.4 ±18.8 |
|  | $F_{cat}$ | 90.4 ±11.5 | 84.0 ±22.1 | 88.0 ±19.7 | 28.9 ±26.2 | 17.6 ±18.8 |
|  | $F_{PM}$ | 81.2 ±15.3 | 67.7 ±28.7 | 82.1 ±22.2 | 40.2 ±28.0 | 23.1 ±20.3 |
| $D_{IRW}$ | $F_L$ | 92.6 ±8.0 | 88.5 ±17.7 | 86.3 ±19.7 | 37.8 ±27.3 | 23.6 ±20.4 |
|  | $F_{L+1}$ | 82.4 ±14.0 | 77.2 ±24.0 | 72.1 ±29.8 | 68.5 ±29.5 | 38.0 ±25.3 |
|  | $F_{[L,L+1]}$ | 95.5 ±10.0 | 91.2 ±15.0 | 94.1 ±29.0 | 23.5 ±31.5 | 13.7 ±15.3 |
|  | $F_{L \oplus L+1}$ | 95.9 ±10.0 | 91.0 ±20.0 | 94.0 ±11.0 | 15.5 ±20.5 | 13.0 ±16.0 |
|  | $F_{cat}$ | 96.1 ±4.9 | 91.8 ±14.0 | 94.1 ±11.4 | 19.1 ±21.6 | 14.1 ±16.2 |
| TRUSTED | $F_{PM}$ | **97.0** ±**4.0** | **93.2** ±**11.5** | **95.1** ±**10.0** | **15.4** ±**19.2** | **11.7** ±**13.7** |

available from the training set (*i.e.*, $D_M$ and $D_{IRW}$) achieve stronger results than those relying on output of softmax scores solely (*i.e.*, $E$ and MSP).

Interestingly, we observe that $D_M$ achieves the best results when relying on the last layer solely (*i.e.*, using $F_{L+1}$). Considering additional layers through concatenation or mean hurts the performances of $D_M$. This is not the case when relying on $D_{IRW}$. Indeed layer aggregation improves the performance of the detector demonstrating the relevance of using $D_{IRW}$ over $D_M$ as an OOD score. Relying on Mahalanobis as OOD score suppose that the representation follows a multivariate Gaussian distribution which might be too strong assumption in the case of layer aggregation. On the contrary $D_{IRW}$ do not rely on any distributional assumption.

## 5.2 On the pretrained encoder choice

Tab. 3: Average (over 480 model configurations) performance per pretrained encoder type.

| Model | Score | Aggregation | AUROC | AUPR-IN | AUPR-OUT | FPR | Err |
|---|---|---|---|---|---|---|---|
| BERT | MSP | Soft. | 89.6 ±9.3 | 84.1 ±19.8 | 80.8 ±25.5 | 46.4 ±30.8 | 25.8 ±22.4 |
|  | $E$ | $F_{L+1}$ | 89.7 ±9.9 | 85.2 ±19.2 | 79.9 ±28.2 | 45.5 ±34.4 | 23.5 ±22.2 |
|  | $D_M$ | $F_L$ | 95.9 ±6.9 | 91.9 ±17.3 | 93.1 ±15.4 | 15.9 ±21.5 | 10.7 ±15.1 |
|  |  | $F_{L+1}$ | 70.7 ±13.0 | 51.9 ±31.5 | 74.3 ±26.9 | 62.3 ±22.2 | 37.4 ±22.1 |
|  |  | $F_{cat}$ | 92.2 ±8.8 | 81.9 ±24.4 | 92.2 ±12.4 | 29.3 ±26.5 | 21.5 ±21.6 |
|  |  | $F_{PM}$ | 80.5 ±16.0 | 65.9 ±30.3 | 81.9 ±21.8 | 42.1 ±28.9 | 24.9 ±21.6 |
|  | $D_{IRW}$ | $F_L$ | 92.6 ±7.7 | 88.7 ±18.2 | 87.0 ±19.1 | 38.0 ±27.8 | 23.8 ±20.4 |
|  |  | $F_{L+1}$ | 81.1 ±14.7 | 76.6 ±24.8 | 71.7 ±29.6 | 72.3 ±29.4 | 40.8 ±25.9 |
|  |  | $F_{cat}$ | 96.5 ±5.2 | 92.1 ±15.7 | 95.8 ±9.2 | 15.9 ±22.1 | 12.8 ±17.9 |
|  | TRUSTED | $F_{PM}$ | **97.4** ±4.1 | **93.6** ±12.8 | **96.4** ±8.6 | **12.6** ±19.6 | **10.4** ±15.6 |
| DIS. | MSP | Soft. | 88.2 ±9.7 | 82.7 ±20.4 | 77.2 ±27.9 | 51.6 ±30.4 | 28.0 ±22.7 |
|  | $E$ | $F_{L+1}$ | 88.1 ±10.9 | 83.3 ±20.8 | 77.1 ±29.4 | 50.3 ±36.1 | 26.2 ±24.1 |
|  | $D_M$ | $F_L$ | 94.1 ±9.0 | 89.4 ±20.2 | 90.6 ±18.0 | 21.6 ±24.3 | 13.8 ±18.1 |
|  |  | $F_{L+1}$ | 72.3 ±13.9 | 55.7 ±33.1 | 72.3 ±29.3 | 63.8 ±23.1 | 37.8 ±24.0 |
|  |  | $F_{cat}$ | 89.2 ±11.6 | 83.9 ±21.7 | 85.6 ±21.5 | 30.8 ±25.7 | 17.4 ±18.2 |
|  |  | $F_{PM}$ | 80.0 ±15.6 | 68.6 ±28.4 | 79.2 ±24.7 | 43.8 ±28.1 | 24.1 ±20.8 |
|  | $D_{IRW}$ | $F_L$ | 91.2 ±9.2 | 86.5 ±19.9 | 84.6 ±21.5 | 43.0 ±28.1 | 26.8 ±21.9 |
|  |  | $F_{L+1}$ | 78.4 ±15.4 | 73.5 ±25.5 | 67.8 ±32.1 | 76.1 ±26.3 | 41.7 ±25.9 |
|  |  | $F_{cat}$ | 96.3 ±3.9 | 91.5 ±13.3 | 94.0 ±11.7 | 18.9 ±20.7 | 14.3 ±14.9 |
|  | TRUSTED | $F_{PM}$ | **97.3** ±2.9 | **93.3** ±10.4 | **95.1** ±9.9 | **14.1** ±17.1 | **11.1** ±11.5 |
| ROB. | MSP | Soft. | 91.4 ±7.9 | 85.9 ±16.4 | 83.0 ±23.0 | 39.1 ±26.0 | 22.6 ±18.7 |
|  | $E$ | $F_{L+1}$ | 91.7 ±8.0 | 86.2 ±16.8 | 82.6 ±24.1 | 39.2 ±28.5 | 22.0 ±19.5 |
|  | $D_M$ | $F_L$ | 91.7 ±11.9 | 86.6 ±22.0 | 90.9 ±15.5 | 21.6 ±24.7 | 13.5 ±17.5 |
|  |  | $F_{L+1}$ | 72.1 ±14.0 | 56.1 ±31.4 | 73.4 ±28.8 | 61.7 ±23.8 | 35.9 ±22.4 |
|  |  | $F_{cat}$ | 89.8 ±13.1 | 85.9 ±20.0 | 86.5 ±22.5 | 26.8 ±26.4 | 14.4 ±15.8 |
|  |  | $F_{PM}$ | 82.8 ±14.4 | 68.4 ±27.4 | 85.0 ±19.7 | 35.1 ±26.5 | 20.5 ±18.2 |
|  | $D_{IRW}$ | $F_L$ | 93.8 ±6.6 | 90.2 ±14.7 | 87.4 ±18.2 | 32.9 ±25.3 | 20.5 ±18.4 |
|  |  | $F_{L+1}$ | 87.2 ±10.0 | 81.0 ±21.0 | 76.4 ±27.2 | 58.1 ±29.4 | 32.2 ±23.2 |
|  |  | $F_{cat}$ | 95.7 ±5.5 | 91.9 ±13.1 | 92.6 ±12.6 | 22.1 ±21.8 | 15.2 ±15.6 |
|  | TRUSTED | $F_{PM}$ | **96.3** ±4.7 | **92.7** ±11.3 | **93.9** ±11.2 | **19.1** ±20.2 | **13.4** ±13.8 |

When training and deploying a classifier, a key question is choosing a pretrained encoder. It can be beneficial in critical applications to trade off the main task accuracy to ensure better OOD detection. In Tab. 3, we report the individual performance of OOD methods on three types of classifiers. Although `TRUSTED` achieves state-of-the-art on all configurations, it is worth noticing a difference in performance concerning the type of pretrained model. It is also important to remark that for `ROB` both MSP and $E$ achieve on-par performances with $D_M$ while not requiring any extra training information. Overall, for a given method (*i.e.* $D_M$ or $D_{IRW}$), the ranking of detectors performances according to the type of feature extractor remain still. This validates the use of the mean-aggregation procedure of `TRUSTED`. Overall, based on the difference of OOD detection performance of Tab. 3, we recommend to use `TRUSTED` on `BERT` or `DIS.` if accurate OOD detection is required.

## 5.3 Impact of the training dataset

OOD detection performance depends on the nature of what is considered in-distribution (the training distribution in our case). Thus, it is interesting to study the performance per-`IN-DS` as reported in Tab. 4. Even though `TRUSTED` achieves strong results in terms of `AUROC`, we observe a high `FPR` on SST2. From Fig. 4, we observe that IMDB and SST2 are harder to detect, especially for $D_{\mathrm{M}}$. Finally, since high `AUROC` does not necessarily imply a low `FPR`, it is crucial to take both into account when designing an OOD detector.

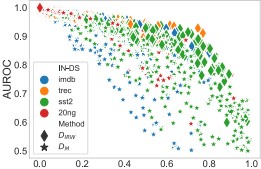

Fig. 4: `AUROC`&`FPR` trade-off.

| IN-DS | Score | AUROC | AUPR-IN | AUPR-OUT | FPR | Err |
|---|---|---|---|---|---|---|
| 20ng | TRUSTED | **98.4** ±1.8 | **96.8** ±4.8 | **98.0** ±4.2 | **8.0** ±10.5 | **6.4** ±6.8 |
| | $D_{\mathrm{M}}$ | 97.6 ±4.6 | 95.1 ±9.9 | 97.4 ±6.4 | 10.1 ±13.6 | 7.6 ±7.5 |
| imdb | TRUSTED | **98.6** ±2.1 | **99.8** ±0.4 | **88.6** ±15.5 | **8.0** ±13.9 | **5.2** ±2.0 |
| | $D_{\mathrm{M}}$ | 93.3 ±9.8 | 98.0 ±5.4 | 77.3 ±24.7 | 19.3 ±20.6 | 6.4 ±2.8 |
| sst2 | TRUSTED | **93.8** ±5.8 | **86.0** ±17.2 | **93.9** ±9.4 | **30.7** ±22.9 | **22.1** ±17.9 |
| | $D_{\mathrm{M}}$ | 86.3 ±12.3 | 71.7 ±29.8 | 90.5 ±12.9 | 43.0 ±25.2 | 30.4 ±22.9 |
| trec | TRUSTED | 97.6 ±2.3 | 91.8 ±8.1 | 99.3 ±1.1 | 12.2 ±15.3 | 11.0 ±12.7 |
| | $D_{\mathrm{M}}$ | **99.0** ±1.2 | **94.9** ±6.8 | **99.8** ±0.4 | **4.4** ±7.4 | **4.3** ±6.2 |

Tab. 4: Average OOD detection performance per `IN-DS`.

## 6 Dynamic Experimental Results

Most OOD detection methods are tested on specific checkpoints, where the selection criterion is often unclear. The consequences of this selection on OOD detection are rarely studied. This section aims to respond to this by measuring the OOD detection performance of methods on various checkpoint finetuning of the pretrained encoder. We will use 5 different checkpoints taken after 1k, 3k, 5k, 10k, 15k, and 20k iterations. Training curves of the models are given in Fig. 7. Notice that after 3k iterations models have converged and no over-fitting is observed even after 20k iterations (*i.e.,* we do not observe an increase in validation loss).

**Overall analysis.** We report the results of the dynamic analysis on the various pretrained models in Fig. 5. For all the methods and models (except $D_{\mathrm{M}}$ on `ROB`), we observe that training longer the classifier hurts detection. Interestingly, this drop in performance has a higher impact on `FPR` compared to `AUROC`. Thus, it *is better to use an early stopping criterion to ensure proper OOD detection performance.* In addition, we observed that `TRUSTED` (corresponding to $D_{\mathrm{IRW}}$) achieves better detection results and that $D_{\mathrm{M}}$ outperforms `TRUSTED` for checkpoints larger than 10k on `ROB`.

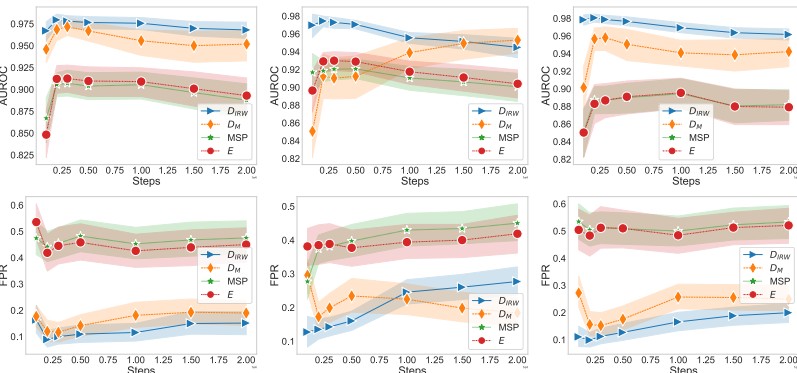

Fig. 5: Detection performance of different pretrained encoder during finetuning. First column correspond to `BERT`, second to `ROB` and last to `DIS`.

**Analysis Per In-Dataset.** Fig. 6 reports the results of the dynamical analysis per IN-DS. We observe that $D_{\text{IRW}}$ is consistently better than $D_{\text{M}}$ with sensible improvement on IMDB and SST2, which are the hardest benchmarks. We observe a similar trend to the previous experiment: *training longer the classifiers hurt their OOD detection performances*. Similar observations hold for FPR, AUPR-IN, and AUPR-OUT that are postponed to the Supplementary Material (see Fig. 10).

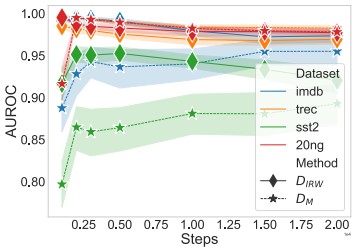

Fig. 6: AUROC per IN-DS during fine-tuning.

## 7 Conclusions and Future Directions

In this work, we introduced TRUSTED a novel OOD detector that relies on information available in all the hidden layers of a network. TRUSTED leverages a novel similarity score built on top of the Integrate Rank-Weighted depth. We conduct extensive numerical experiments proving that it consistently outperforms previous approaches, including detection based on the Mahalanobis distance. Our comprehensive evaluation framework demonstrates that, in general, OOD performances vary depending on several hyperparameters of the models, the datasets, and the detector's feature extraction step. Thus, we would like to promote the use of such exhaustive evaluation frameworks for future search to assess AI systems' safety tools properly. Another interesting question is the detection inference-time / accuracy trade-off, which is instrumental for the practitioner.

## Acknowledgments

This work was also granted access to the HPC resources of IDRIS under the allocation 2021-AP010611665 as well as under the project 2021-101838 made by GENCI. This work has been supported by the project PSPC AIDA: 2019-PSPC-09 funded by BPI-France.

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
