\left(\Pr\left(s(\mathbf{X}) > \gamma \mid Z = 0\right), \Pr\left(s(\mathbf{X}) \leq \gamma \mid Z = 1\right)\right)$, which plots the true detection rates against the false alarm rates. The `AUROC` corresponds to the probability that an in-distribution example $\mathbf{X}_{in}$ has higher score than an OOD sample $\mathbf{X}_{out}$: $\texttt{AUROC} = \Pr(s(\mathbf{X}_{in}) > s(\mathbf{X}_{out}))$, as can be checked from elementary computations.

**Area Under the Precision-Recall curve (`AUPR-IN`/`AUPR-OUT`) [31].** It is the area under the precision-recall curve $\gamma \mapsto \left(\Pr\left(Z = 1 \mid s(\mathbf{X}) \leq \gamma\right), \Pr\left(s(\mathbf{X}) \leq \gamma \mid Z = 1\right)\right)$

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

 | **98.4** $_{\pm1.8}$ | **96.8** $_{\pm4.8}$ | **98.0** $_{\pm4.2}$ | **8.0** $_{\pm10.5}$ | **6.4** $_{\pm6.8}$ |
| | $D_M$ | 97.6 $_{\pm4.6}$ | 95.1 $_{\pm9.9}$ | 97.4 $_{\pm6.4}$ | 10.1 $_{\pm13.6}$ | 7.6 $_{\pm7.5}$ |
| imdb | TRUSTED | **98.6** $_{\pm2.1}$ | **99.8** $_{\pm0.4}$ | **88.6** $_{\pm15.5}$ | **8.0** $_{\pm13.9}$ | **5.2** $_{\pm2.0}$ |
| | $D_M$ | 93.3 $_{\pm9.8}$ | 98.0 $_{\pm5.4}$ | 77.3 $_{\pm24.7}$ | 19.3 $_{\pm20.6}$ | 6.4 $_{\pm2.8}$ |
| sst2 | TRUSTED | **93.8** $_{\pm5.8}$ | **86.0** $_{\pm17.2}$ | **93.9** $_{\pm9.4}$ | **30.7** $_{\pm22.9}$ | **22.1** $_{\pm17.9}$ |
| | $D_M$ | 86.3 $_{\pm12.3}$ | 71.7 $_{\pm29.8}$ | 90.5 $_{\pm12.9}$ | 43.0 $_{\pm25.2}$ | 30.4 $_{\pm22.9}$ |
| trec | TRUSTED | 97.6 $_{\pm2.3}$ | 91.8 $_{\pm8.1}$ | 99.3 $_{\pm1.1}$ | 12.2 $_{\pm15.3}$ | 11.0 $_{\pm12.7}$ |
| | $D_M$ | **99.0** $_{\pm1.2}$ | **94.9** $_{\pm6.8}$ | **99.8** $_{\pm0.4}$ | **4.4** $_{\pm7.4}$ | **4.3** $_{\pm6.2}$ |

Fig. 4: `AUROC`&`FPR` trade-off.          Tab. 4: Average OOD detection performance per `IN-DS`.

## 6 Dynamic Experimental Results

Most OOD detection methods are tested on specific checkpoints, where the selection criterion is often unclear. The consequences of this selection on OOD detection are rarely studied. This section aims to respond to this by measuring the OOD detection performance of methods on various checkpoint finetuning of the pretrained encoder. We will use 5 different checkpoints taken after 1k, 3k, 5k, 10k, 15k, and 20k iterations. Training curves of the models are given in Fig. 7. Notice that after 3k iterations models have converged and no over-fitting is observed even after 20k iterations (*i.e.,* we do not observe an increase in validation loss).

**Overall analysis.** We report the results of the dynamic analysis on the various pretrained models in Fig. 5. For all the methods and models (except $D_M$ on `ROB`), we observe that training longer the classifier hurts detection. Interestingly, this drop in performance has a higher impact on `FPR` compared to `AUROC`. Thus, it *is better to use an early stopping criterion to ensure proper OOD detection performance.* In addition, we observed that `TRUSTED` (corresponding to $D_{IRW}$) achieves better detection results and that $D_M$ outperforms `TRUSTED` for checkpoints larger than 10k on `ROB`.

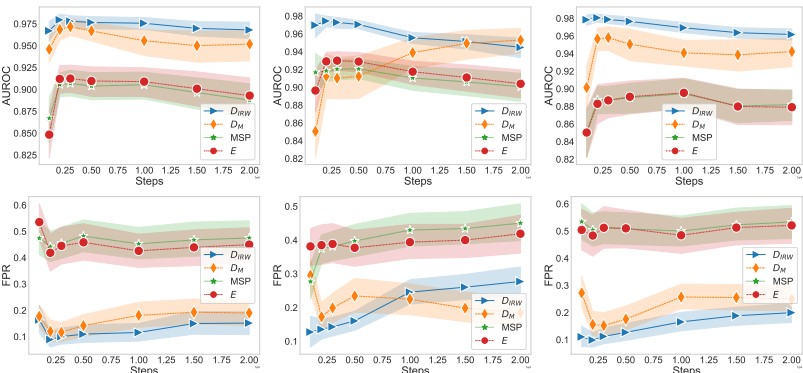

Fig. 5: Detection performance of different pretrained encoder during finetuning. First column correspond to `BERT`, second to `ROB` and last to `DIS`.

**Analysis Per In-Dataset.** Fig. 6 reports the results of the dynamical analysis per IN-DS. We observe that $D_{\mathrm{IRW}}$ is consistently better than $D_{\mathrm{M}}$ with sensible improvement on IMDB and SST2, which are the hardest benchmarks. We observe a similar trend to the previous experiment: *training longer the classifiers hurt their OOD detection performances*. Similar observations hold for FPR, AUPR-IN, and AUPR-OUT that are postponed to the Supplementary Material (see Fig. 10).

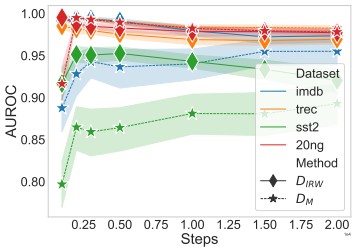

Fig. 6: AUROC per IN-DS during fine-tuning.

## 7 Conclusions and Future Directions

In this work, we introduced TRUSTED a novel OOD detector that relies on information available in all the hidden layers of a network. TRUSTED leverages a novel similarity score built on top of the Integrate Rank-Weighted depth. We conduct extensive numerical experiments proving that it consistently outperforms previous approaches, including detection based on the Mahalanobis distance. Our comprehensive evaluation framework demonstrates that, in general, OOD performances vary depending on several hyperparameters of the models, the datasets, and the detector's feature extraction step. Thus, we would like to promote the use of such exhaustive evaluation frameworks for future search to assess AI systems' safety tools properly. Another interesting question is the detection inference-time / accuracy trade-off, which is instrumental for the practitioner.

## Acknowledgments

This work was also granted access to the HPC resources of IDRIS under the allocation 2021-AP010611665 as well as under the project 2021-101838 made by GENCI. This work has been supported by the project PSPC AIDA: 2019-PSPC-09 funded by BPI-France.

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

# Appendices

## A  Experimental Details

In this section we gather additional experimental details. For completeness we provide the used algorithms for both TRUSTED and $D_{\mathrm{IRW}}$ (see Sec. A.1), we also gather additional benchmarks details (see Sec. A.2), hyperparameter used during training (see Sec. A.3) as well as the training curves (see Sec. A.4).

### A.1  Algorithms

In this part, we present algorithms to compute $D_{\mathrm{IRW}}$ (see Algorithm 1) and TRUSTED (see Algorithm 2).

---

**Algorithm 1** Approximation of the IRW depth

---

*Initialization:* test sample $x$, $n_{\mathrm{proj}}$, $\mathbf{X} = [x_1, \ldots, x_n]^{\top}$.

  1: Construct $\mathbf{U} \in \mathbb{R}^{d \times n_{\mathrm{proj}}}$ by sampling uniformly $n_{\mathrm{proj}}$ vectors $U_1, \ldots, U_{n_{\mathrm{proj}}}$ in $\mathbb{S}^{d-1}$

  2: Compute $\mathbf{M} = \mathbf{X}\mathbf{U}$ and $x^{\top}\mathbf{U}$

  3: Compute the rank value $\sigma(j)$, the rank of $x^{\top}\mathbf{U}$ in $\mathbf{M}_{:,j}$ for every $j \leq n_{\mathrm{proj}}$

  4: Set $D = \frac{1}{n_{\mathrm{proj}}} \sum_{j=1}^{n_{\mathrm{proj}}} \sigma(j)$

    **Output**: $\widetilde{D}_{\mathrm{IRW}}(x, \mathbf{X}) = D$

---

| Dataset | #train | #dev | #test | #class |
|---------|--------|------|-------|--------|
| SST2 | 67349 | 872 | 1821 | 2 |
| IMDB | 22500 | 2500 | 25000 | 2 |
| TREC10 | 4907 | 545 | 500 | 6 |
| 20NG | 15056 | 1876 | 1896 | 20 |
| MNLI | - | - | 19643 | - |
| RTE | - | - | 3000 | - |
| Multi30K | - | - | 2532 | - |
| WMT16 | - | - | 2999 | - |

Tab. 5: Statistics of the considered benchmark.

---

**Algorithm 2** Computation of TRUSTED

---

*Initialization:* $\mathbf{x}, n_{\mathrm{proj}}, \mathcal{S}_n, \hat{y}$.

1: Compute $F_{\mathrm{PM}}(\mathbf{x})$
2: **for** $y = 1, \ldots, C$ **do**
    Compute $F_{\mathrm{PM}}(\mathcal{S}_{n,y}^{\mathrm{train}})$
3: **end for**
4: Compute $D_{\mathrm{IRW}}(F_{\mathrm{PM}}(\mathbf{x}), F_{\mathrm{PM}}(\mathcal{S}_{n,\hat{y}}^{\mathrm{train}}))$ using Algorithm 1 with $F_{\mathrm{PM}}(\mathbf{x})$, $n_{\mathrm{proj}}$ and $F_{\mathrm{PM}}(\mathcal{S}_{n,\hat{y}}^{\mathrm{train}})$

   **Output**: $s_{\text{TRUSTED}}(\mathbf{x}) = D_{\mathrm{IRW}}(F_{\mathrm{PM}}(\mathbf{x}), F_{\mathrm{PM}}(\mathcal{S}_{n,\hat{y}}^{\mathrm{train}}))$

---

## A.2 Benchmark Details

We report in Tab. 5 statistics related to the datasets of our benchmark. The only difference between our work and the one from [103] is that we considered the pair IMDB and SST2 a valid OOD pair of IN-DS/OOD-DS as this can be seen as a background shift (see [78]).

**Remark 1** *We initially started to work with the appealing benchmark introduced by [60] which introduces an alternative standard that addresses the limitation of the benchmark proposed by [100] (e.g. there is no category overlap between training and OOD test examples in the non-semantic shift dataset). Additionally, [60] put effort into designing a dataset that can categorize shifts as belonging to semantic or background shift [2, 78]. However, we failed to reproduce the baseline results from [60] even after contacting the authors.*

## A.3 Training Parameters

In this section, the detail the main hyper-parameters that were used for finetuning the pretrained encoders. It is worth noting that we use the same set of hyperparameters for all the different encoders [26, 24**?** ]. The dropout rate [89] is set to $0.2$ [], we train with a batch size of 32, we use ADAMW [55, 69, 4]. Additionally, we set the weight decay to $0.01$, the warmup ratio to $0.06$, and the learning rate to $10^{-5}$. All the models were trained during 20k iterations with different seeds.

## A.4 Training Curves

In order to understand the change of performance while finetuning the model, it is crucial to understand when and if the different models have converged. Thus we report in Fig. 7 dev losses and dev and test accuracy. From Fig. 7, we can observe that a pretrained encoder finetuned on 20ng takes more time to converge compared to the same pretrained encoder finetuned on either SST2, IMDB, or trec. Additionally, after 2k updates, both dev and test accuracy are stable on SST2, IMDB, and trec, while for 20ng, it requires 6k steps. Last, it is worth noting (see Tab. 1) that ROB. achieves the best accuracy overall. BERT is the second best and achieves stronger test accuracy than DIS. on the different datasets.

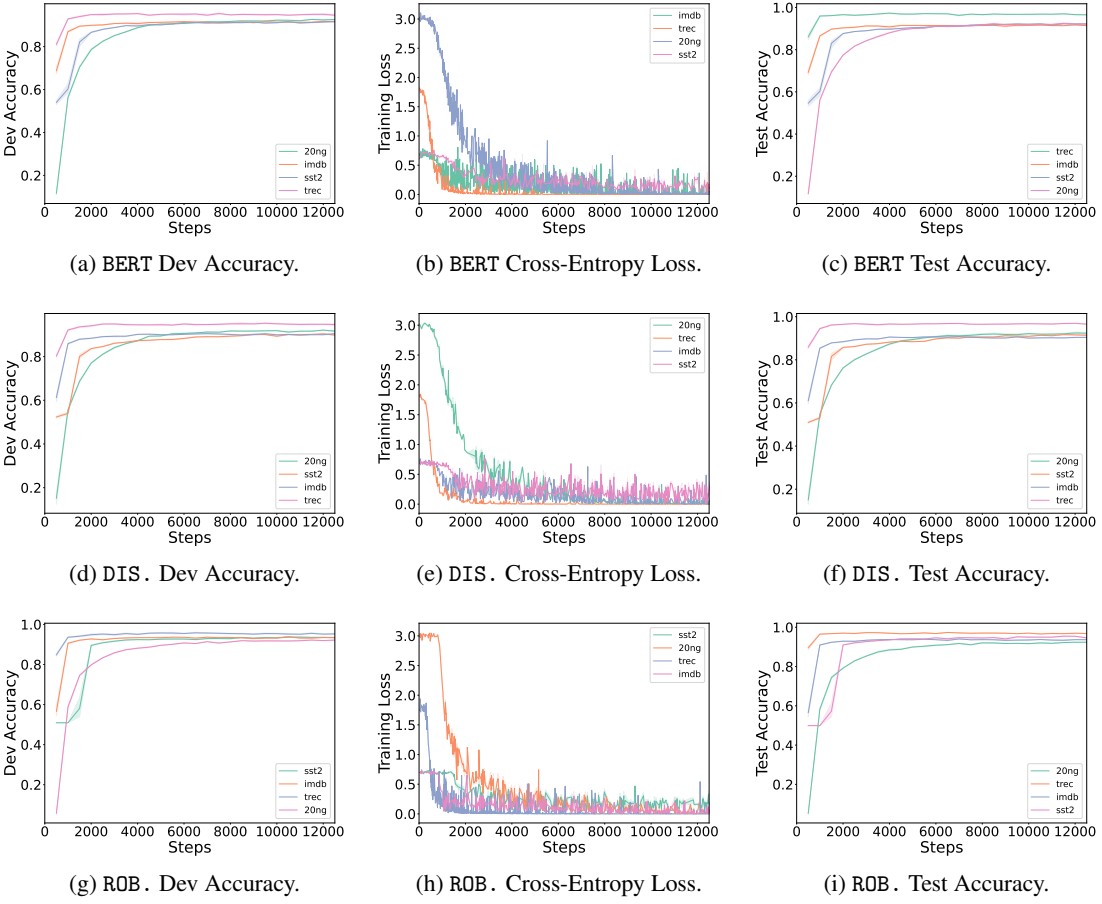

Fig. 7: Dev and Test curves obtained during finetuning of the pretrained transformers on different datasets for the three considered seeds.

# B    Additional Static Experimental Results

In this section, we report additional static experiment results. Formally, we aim to gain understanding:

- on the role of the IN-DS (see Sec. B.1) and on the impact of the choice of the pretrained encoder [27].
- on the different trade-off that exists between the different metrics (see Sec. B.2)
- on the impact of performance of the detection methods depending on the OOD distribution (see Sec. B.3).

## B.1    Analysis Per In-Dataset

We present in Tab. 6 the average performance per IN-DS. On 3 out of 4 datasets TRUSTED achieves the best results and outperforms other methods. Interestingly, the table shows the key importance of the training corpus. As an example, we observe that detection methods applied to classifiers trained on sst2 are less efficient (by several AUROC points) compared to other IN-DS.
A finer analysis of this phenomenon can be conducted using Tab. 7. From Tab. 7, we observe that this phenomenon is consistent accross all the pretrained classifiers (*i.e.,* BERT, ROB. and DIS.). We observe that different TRUSTED does not work uniformly better on all pretrained models. For example, the best results on 20ng are obtained for BERT while on sst2 it is obtained for DIS..
**Takeaways:** Both IN-DS and pretrained model choices are essential to ensure good detection performance.

Tab. 6: Average OOD detection performance (in %) per IN-DS.

| Score | Method | AUROC | AUPR-IN | AUPR-OUT | FPR | Err |
|---|---|---|---|---|---|---|
| 20ng | TRUSTED | **98.4** $\pm1.8$ | **96.8** $\pm4.8$ | **98.0** $\pm4.2$ | **8.0** $\pm10.5$ | **6.4** $\pm6.8$ |
| | $D_{\mathrm{M}}$ | 97.6 $\pm4.6$ | 95.1 $\pm9.9$ | 97.4 $\pm6.4$ | 10.1 $\pm13.6$ | 7.6 $\pm7.5$ |
| | $E$ | 94.9 $\pm3.7$ | 88.4 $\pm10.2$ | 95.4 $\pm6.5$ | 21.3 $\pm17.1$ | 14.8 $\pm11.5$ |
| | MSP | 92.6 $\pm4.8$ | 85.0 $\pm12.0$ | 93.5 $\pm8.3$ | 31.1 $\pm16.3$ | 20.8 $\pm11.0$ |
| imdb | TRUSTED | **98.6** $\pm2.1$ | **99.8** $\pm0.4$ | **88.6** $\pm15.5$ | **8.0** $\pm13.9$ | **5.2** $\pm2.0$ |
| | $D_{\mathrm{M}}$ | 93.3 $\pm9.8$ | 98.0 $\pm5.4$ | 77.3 $\pm24.7$ | 19.3 $\pm20.6$ | 6.4 $\pm2.8$ |
| | $E$ | 87.7 $\pm9.0$ | 97.9 $\pm2.7$ | 40.0 $\pm24.2$ | 64.4 $\pm27.3$ | 12.8 $\pm8.9$ |
| | MSP | 89.7 $\pm7.5$ | 98.2 $\pm2.1$ | 43.3 $\pm22.5$ | 56.2 $\pm22.5$ | 12.0 $\pm7.8$ |
| sst2 | TRUSTED | **93.8** $\pm5.8$ | **86.0** $\pm17.2$ | **93.9** $\pm9.4$ | **30.7** $\pm22.9$ | **22.1** $\pm17.9$ |
| | $D_{\mathrm{M}}$ | 86.3 $\pm12.3$ | 71.7 $\pm29.8$ | 90.5 $\pm12.9$ | 43.0 $\pm25.2$ | 30.4 $\pm22.9$ |
| | $E$ | 80.8 $\pm10.0$ | 70.6 $\pm25.9$ | 81.9 $\pm17.0$ | 76.2 $\pm18.9$ | 50.1 $\pm21.7$ |
| | MSP | 81.2 $\pm10.0$ | 71.3 $\pm25.7$ | 82.6 $\pm16.1$ | 75.8 $\pm17.3$ | 50.1 $\pm21.4$ |
| trec | TRUSTED | 97.6 $\pm2.3$ | 91.8 $\pm8.1$ | 99.3 $\pm1.1$ | 12.2 $\pm15.3$ | 11.0 $\pm12.7$ |
| | $D_{\mathrm{M}}$ | **99.0** $\pm1.2$ | **94.9** $\pm6.8$ | **99.8** $\pm0.4$ | **4.4** $\pm7.4$ | **4.3** $\pm6.2$ |
| | $E$ | 96.9 $\pm3.3$ | 85.6 $\pm13.9$ | 99.3 $\pm1.0$ | 15.5 $\pm15.3$ | 13.9 $\pm12.7$ |
| | MSP | 96.2 $\pm3.4$ | 85.1 $\pm13.7$ | 99.1 $\pm1.2$ | 16.9 $\pm15.8$ | 15.2 $\pm13.2$ |

## B.2   Trade-off between metrics

We report Fig. 8 the different trade-off that exist between the considered metrics. Interestingly, a high AUROC does not imply a low FPR. Similar conclusions can be drawn regarding AUPR-IN and AUPR-OUT.

**Takeaways.** Fig. 8 illustrates that all metrics matters and should be considered when comparing detection methods.

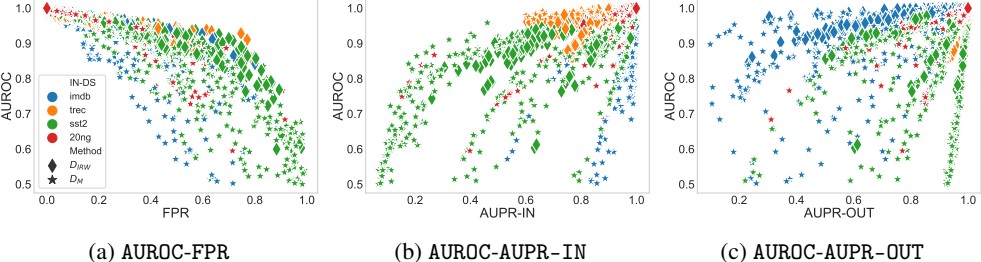

(a) AUROC-FPR          (b) AUROC-AUPR-IN          (c) AUROC-AUPR-OUT

Fig. 8: Trade-off between different metrics for all considered configurations.

## B.3   Analysis Per Out-Dataset

We report on Tab. 8 the detailed results of the detection methods for different OUT-DS. It is worth noting that TRUSTED achieves best results on 22 configurations.

**Takeaways.** Tab. 8 illustrates the impact of the type of OOD sample on the detection performances.

## C   Additional Dynamical Experimental Results

In this section, we gather additional dynamical analysis [25, 23]. We further illustrate the importance of dynamical probing in Sec. C.1. We then conduct a dynamical study of the impact of the OOD dataset in Sec. C.2. In Sec. C.3, we study the role of the pretrained encoder and in Sec. C.4. Last, we gather per encoder, per IN-DS and per OUT-DS analysis.

Tab. 7: Average OOD detection performance (in %) per IN-DS and per pretrained encoders.

| IN DS | MODEL | TECH | AUROC | AUPR-IN | AUPR-OUT | FPR | Err |
|---|---|---|---|---|---|---|---|
| 20ng | BERT | TRUSTED | **99.9** $\pm0.3$ | **99.4** $\pm1.4$ | **100.0** $\pm0.0$ | **0.4** $\pm0.9$ | **1.0** $\pm1.0$ |
| | | $D_M$ | 98.8 $\pm1.7$ | 97.6 $\pm2.6$ | 98.4 $\pm4.6$ | 6.1 $\pm8.7$ | 5.1 $\pm3.8$ |
| | | $E$ | 95.4 $\pm3.8$ | 89.0 $\pm10.1$ | 95.9 $\pm6.1$ | 21.0 $\pm20.0$ | 15.0 $\pm13.6$ |
| | | MSP | 93.7 $\pm4.1$ | 86.4 $\pm11.7$ | 94.8 $\pm6.0$ | 28.1 $\pm15.6$ | 19.5 $\pm11.1$ |
| | Dist | TRUSTED | **99.0** $\pm0.7$ | **97.8** $\pm2.6$ | **99.0** $\pm1.4$ | **3.5** $\pm4.0$ | **4.3** $\pm2.7$ |
| | | $D_M$ | 97.7 $\pm4.2$ | 96.0 $\pm5.6$ | 97.0 $\pm9.0$ | 10.9 $\pm16.7$ | 7.8 $\pm8.1$ |
| | | $E$ | 96.4 $\pm3.0$ | 91.2 $\pm7.8$ | 96.9 $\pm4.8$ | 15.7 $\pm14.9$ | 11.6 $\pm10.4$ |
| | | MSP | 93.9 $\pm4.1$ | 86.8 $\pm11.1$ | 95.2 $\pm5.6$ | 25.6 $\pm15.0$ | 17.8 $\pm10.5$ |
| | Rob | TRUSTED | **96.8** $\pm1.7$ | **94.0** $\pm6.0$ | **95.6** $\pm5.8$ | **17.2** $\pm10.7$ | **12.1** $\pm7.0$ |
| | | $D_M$ | 96.5 $\pm5.9$ | 92.3 $\pm14.2$ | 96.9 $\pm5.4$ | 12.9 $\pm13.8$ | 9.4 $\pm8.7$ |
| | | $E$ | 93.5 $\pm3.6$ | 85.9 $\pm11.2$ | 94.0 $\pm7.5$ | 25.3 $\pm14.6$ | 16.8 $\pm10.0$ |
| | | MSP | 90.7 $\pm5.1$ | 82.7 $\pm12.4$ | 91.4 $\pm10.6$ | 37.3 $\pm15.8$ | 23.9 $\pm10.7$ |
| imdb | BERT | TRUSTED | **98.9** $\pm1.8$ | **99.9** $\pm0.2$ | **91.1** $\pm14.8$ | **5.6** $\pm11.1$ | **4.7** $\pm1.0$ |
| | | $D_M$ | 96.5 $\pm5.4$ | 99.5 $\pm0.8$ | 80.9 $\pm24.3$ | 13.9 $\pm18.7$ | 5.5 $\pm1.6$ |
| | | $E$ | 85.6 $\pm9.8$ | 97.6 $\pm3.2$ | 34.4 $\pm22.3$ | 73.8 $\pm21.1$ | 13.9 $\pm9.5$ |
| | | MSP | 89.2 $\pm7.0$ | 98.1 $\pm2.3$ | 41.3 $\pm21.8$ | 60.5 $\pm20.3$ | 12.6 $\pm8.3$ |
| | Dist | TRUSTED | **98.8** $\pm1.8$ | **99.8** $\pm0.3$ | **89.2** $\pm13.2$ | **6.2** $\pm10.5$ | **4.9** $\pm1.2$ |
| | | $D_M$ | 95.0 $\pm6.6$ | 99.2 $\pm1.0$ | 75.9 $\pm26.8$ | 20.2 $\pm21.3$ | 6.3 $\pm2.4$ |
| | | $E$ | 86.2 $\pm9.1$ | 97.7 $\pm2.9$ | 38.6 $\pm25.6$ | 69.1 $\pm27.5$ | 13.3 $\pm9.3$ |
| | | MSP | 88.0 $\pm7.8$ | 98.0 $\pm2.3$ | 39.0 $\pm21.6$ | 64.1 $\pm17.6$ | 12.9 $\pm8.2$ |
| | Rob | TRUSTED | **98.2** $\pm2.7$ | **99.7** $\pm0.6$ | **86.0** $\pm17.8$ | **12.1** $\pm17.9$ | **5.9** $\pm3.0$ |
| | | $D_M$ | 88.6 $\pm13.3$ | 95.3 $\pm8.5$ | 75.9 $\pm22.5$ | 22.7 $\pm20.7$ | 7.2 $\pm3.7$ |
| | | $E$ | 91.2 $\pm7.2$ | 98.5 $\pm1.8$ | 46.3 $\pm22.9$ | 51.4 $\pm26.8$ | 11.4 $\pm7.8$ |
| | | MSP | 92.0 $\pm7.1$ | 98.6 $\pm1.7$ | 49.6 $\pm22.8$ | 43.9 $\pm23.7$ | 10.6 $\pm6.8$ |
| sst2 | BERT | TRUSTED | **93.2** $\pm5.3$ | **83.4** $\pm19.1$ | **94.4** $\pm7.1$ | **32.0** $\pm23.0$ | **25.1** $\pm20.4$ |
| | | $D_M$ | 90.1 $\pm9.4$ | 78.1 $\pm26.7$ | 91.7 $\pm14.2$ | 36.8 $\pm23.9$ | 25.8 $\pm20.5$ |
| | | $E$ | 81.2 $\pm9.2$ | 70.4 $\pm26.2$ | 82.8 $\pm17.0$ | 75.5 $\pm15.6$ | 49.5 $\pm20.4$ |
| | | MSP | 80.2 $\pm9.5$ | 69.3 $\pm27.0$ | 81.7 $\pm16.0$ | 80.0 $\pm8.9$ | 52.7 $\pm20.2$ |
| | Dist | TRUSTED | **95.1** $\pm3.2$ | **87.5** $\pm14.4$ | **93.8** $\pm11.3$ | **26.2** $\pm18.7$ | **17.8** $\pm11.6$ |
| | | $D_M$ | 86.5 $\pm12.2$ | 71.9 $\pm30.8$ | 91.0 $\pm11.9$ | 43.8 $\pm25.5$ | 31.6 $\pm25.0$ |
| | | $E$ | 76.3 $\pm8.8$ | 65.9 $\pm27.1$ | 77.4 $\pm18.9$ | 86.1 $\pm13.6$ | 55.9 $\pm21.1$ |
| | | MSP | 77.7 $\pm8.5$ | 67.7 $\pm26.5$ | 79.0 $\pm17.7$ | 85.0 $\pm5.9$ | 55.4 $\pm19.6$ |
| | Rob | TRUSTED | **93.2** $\pm7.7$ | **87.2** $\pm17.5$ | **93.6** $\pm9.6$ | **34.0** $\pm25.9$ | **23.5** $\pm19.7$ |
| | | $D_M$ | 82.3 $\pm13.7$ | 65.1 $\pm30.4$ | 88.9 $\pm12.3$ | 48.2 $\pm25.1$ | 33.7 $\pm22.5$ |
| | | $E$ | 84.9 $\pm10.2$ | 75.6 $\pm23.4$ | 85.5 $\pm13.7$ | 67.0 $\pm21.6$ | 44.8 $\pm22.1$ |
| | | MSP | 85.8 $\pm10.3$ | 76.8 $\pm22.5$ | 87.0 $\pm13.3$ | 62.5 $\pm22.6$ | 42.2 $\pm22.2$ |
| trec | BERT | TRUSTED | 98.6 $\pm1.4$ | 95.0 $\pm5.5$ | 99.6 $\pm0.6$ | 7.2 $\pm11.9$ | 6.7 $\pm9.9$ |
| | | $D_M$ | **99.3** $\pm0.7$ | **96.5** $\pm4.6$ | **99.8** $\pm0.2$ | **2.3** $\pm3.7$ | **2.5** $\pm3.1$ |
| | | $E$ | 97.6 $\pm2.6$ | 88.4 $\pm11.7$ | 99.5 $\pm0.7$ | 10.2 $\pm10.3$ | 9.3 $\pm8.5$ |
| | | MSP | 96.9 $\pm2.8$ | 87.4 $\pm11.3$ | 99.3 $\pm0.9$ | 12.6 $\pm11.6$ | 11.4 $\pm9.6$ |
| | Dist | TRUSTED | 97.0 $\pm2.8$ | 89.9 $\pm8.9$ | 99.2 $\pm1.2$ | 16.6 $\pm17.5$ | 14.8 $\pm14.5$ |
| | | $D_M$ | **98.6** $\pm1.6$ | **93.4** $\pm7.5$ | **99.7** $\pm0.5$ | **7.8** $\pm10.6$ | **7.3** $\pm8.8$ |
| | | $E$ | 95.9 $\pm4.1$ | 82.1 $\pm15.6$ | 99.1 $\pm1.3$ | 20.7 $\pm19.1$ | 18.3 $\pm16.0$ |
| | | MSP | 94.9 $\pm4.4$ | 80.8 $\pm16.1$ | 98.7 $\pm1.5$ | 24.3 $\pm19.9$ | 21.5 $\pm16.5$ |
| | Rob | TRUSTED | 97.3 $\pm2.1$ | 90.9 $\pm8.5$ | 99.1 $\pm1.2$ | 12.2 $\pm14.4$ | 11.0 $\pm11.5$ |
| | | $D_M$ | **99.1** $\pm0.9$ | **95.0** $\pm7.4$ | **99.8** $\pm0.3$ | **2.6** $\pm4.1$ | **2.8** $\pm3.4$ |
| | | $E$ | 97.1 $\pm2.5$ | 86.6 $\pm13.2$ | 99.3 $\pm0.9$ | 14.8 $\pm12.7$ | 13.3 $\pm10.4$ |
| | | MSP | 97.0 $\pm2.0$ | 87.5 $\pm11.8$ | 99.3 $\pm0.8$ | 13.3 $\pm11.1$ | 12.0 $\pm9.1$ |

## C.1   On the importance of dynamical probing

In Fig. 9, we report histograms for TRUSTED and Mahanalobis distance. Interestingly, the shape of histograms is changing across checkpoints demonstrating the need for dynamical probing [22, 93].

| OUT DS | MODEL | TECH | Feature Type | AUROC | AUPR-IN | AUPR-OUT | FPR | Err |
|---|---|---|---|---|---|---|---|---|
| 20ng | BERT | TRUSTED | | **97.9** ±2.2 | **99.2** ±0.7 | **87.2** ±19.2 | **11.6** ±14.6 | **4.6** ±4.3 |
| | | $D_M$ | Pooled | 97.4 ±2.6 | 98.2 ±2.2 | 92.0 ±10.3 | 13.6 ±14.8 | 7.7 ±7.7 |
| | | $E$ | Energy | 91.1 ±7.5 | 95.3 ±4.8 | 72.3 ±30.9 | 48.8 ±36.1 | 18.4 ±17.1 |
| | | MSP | softmax | 91.1 ±6.8 | 95.0 ±5.1 | 72.7 ±28.9 | 49.1 ±34.1 | 19.4 ±17.4 |
| | Dist | TRUSTED | | **98.7** ±1.7 | **99.5** ±0.7 | 90.7 ±14.6 | **6.0** ±9.9 | **3.3** ±2.0 |
| | | $D_M$ | Pooled | 97.0 ±4.5 | 97.7 ±5.0 | **92.3** ±11.2 | 13.9 ±16.6 | 7.6 ±7.9 |
| | | $E$ | Energy | 87.7 ±10.1 | 93.7 ±7.6 | 65.7 ±32.4 | 56.9 ±41.0 | 20.7 ±20.3 |
| | | MSP | softmax | 89.1 ±8.3 | 94.3 ±6.2 | 67.9 ±31.0 | 53.1 ±36.6 | 19.9 ±18.7 |
| | Rob | TRUSTED | | **98.6** ±1.5 | **99.1** ±1.1 | **92.9** ±11.5 | **7.7** ±12.3 | **4.7** ±5.4 |
| | | $D_M$ | Pooled | 91.5 ±11.0 | 92.5 ±12.1 | 86.3 ±16.0 | 23.9 ±28.7 | 12.1 ±14.9 |
| | | $E$ | Energy | 95.3 ±4.3 | 97.0 ±4.4 | 79.6 ±24.3 | 30.2 ±27.4 | 12.3 ±12.7 |
| | | MSP | softmax | 94.7 ±6.8 | 96.2 ±8.1 | 79.6 ±23.8 | 28.0 ±24.9 | 11.8 ±12.3 |
| imdb | BERT | TRUSTED | | **95.1** ±5.6 | **80.1** ±21.3 | **99.5** ±0.7 | **21.3** ±26.2 | **20.2** ±24.4 |
| | | $D_M$ | Pooled | 90.2 ±13.4 | 69.5 ±35.8 | 98.9 ±1.7 | 27.9 ±36.0 | 26.3 ±33.6 |
| | | $E$ | Energy | 87.3 ±14.9 | 62.5 ±31.0 | 98.5 ±2.0 | 37.2 ±38.7 | 35.0 ±36.1 |
| | | MSP | softmax | 86.2 ±14.6 | 59.5 ±30.2 | 98.3 ±2.0 | 41.7 ±37.4 | 39.2 ±34.8 |
| | Dist | TRUSTED | | **96.1** ±2.6 | **79.9** ±13.9 | **99.7** ±0.2 | **19.1** ±15.0 | **18.4** ±14.3 |
| | | $D_M$ | Pooled | 86.0 ±17.6 | 64.0 ±38.8 | 98.2 ±2.4 | 36.6 ±42.6 | 34.5 ±39.7 |
| | | $E$ | Energy | 84.9 ±16.6 | 56.1 ±31.5 | 98.0 ±2.6 | 44.5 ±39.7 | 42.0 ±37.0 |
| | | MSP | softmax | 84.7 ±15.3 | 55.2 ±29.6 | 98.1 ±2.3 | 46.6 ±35.9 | 44.0 ±33.3 |
| | Rob | TRUSTED | | **95.3** ±7.4 | **80.6** ±20.7 | **99.5** ±0.9 | **18.3** ±24.1 | **17.5** ±22.5 |
| | | $D_M$ | Pooled | 89.5 ±16.3 | 68.6 ±35.0 | 98.6 ±2.5 | 25.0 ±36.9 | 23.5 ±34.4 |
| | | $E$ | Energy | 89.9 ±12.6 | 64.8 ±23.1 | 98.8 ±1.7 | 35.2 ±34.6 | 33.2 ±32.2 |
| | | MSP | softmax | 90.4 ±11.4 | 65.3 ±21.9 | 98.9 ±1.6 | 34.8 ±32.2 | 32.8 ±30.0 |
| mnli | BERT | TRUSTED | | 95.5 ±6.0 | **82.8** ±18.7 | 99.2 ±0.9 | 19.9 ±26.8 | 19.0 ±24.3 |
| | | $D_M$ | Pooled | **96.5** ±4.0 | 82.3 ±18.8 | **99.3** ±0.9 | **14.8** ±16.2 | **13.7** ±14.8 |
| | | $E$ | Energy | 89.4 ±8.8 | 67.5 ±19.9 | 93.9 ±10.2 | 46.6 ±33.1 | 36.2 ±26.7 |
| | Dist | TRUSTED | | **96.8** ±2.8 | **83.6** ±14.1 | **99.3** ±0.8 | **15.4** ±14.5 | **14.7** ±13.1 |
| | | MSP | softmax | 88.6 ±9.2 | 65.9 ±20.5 | 94.7 ±7.5 | 49.6 ±29.6 | 39.8 ±25.9 |
| | | $D_M$ | Pooled | 94.9 ±6.2 | 76.2 ±24.4 | 98.7 ±1.7 | 19.9 ±19.1 | 17.6 ±17.4 |
| | | $E$ | Energy | 88.4 ±10.4 | 66.0 ±23.3 | 93.1 ±10.7 | 48.6 ±34.4 | 37.2 ±28.1 |
| | | MSP | softmax | 88.3 ±8.9 | 65.2 ±21.4 | 93.8 ±8.1 | 51.4 ±28.5 | 40.1 ±24.8 |
| | Rob | TRUSTED | | **96.2** ±2.9 | **83.0** ±12.3 | **99.0** ±1.7 | **21.4** ±16.7 | **19.2** ±14.8 |
| | | $D_M$ | Pooled | 90.6 ±12.4 | 69.9 ±29.1 | 97.2 ±5.6 | 22.9 ±22.9 | 19.6 ±20.5 |
| | | $E$ | Energy | 91.9 ±6.3 | 69.5 ±18.5 | 95.7 ±6.8 | 41.3 ±25.9 | 32.7 ±21.4 |
| | | MSP | softmax | 91.4 ±6.4 | 69.2 ±17.8 | 96.0 ±5.7 | 42.2 ±23.0 | 34.1 ±20.4 |
| multi30k | BERT | TRUSTED | | **98.6** ±1.6 | **98.0** ±2.1 | **98.9** ±1.5 | 7.3 ±9.3 | **6.5** ±6.0 |
| | | $D_M$ | Pooled | 98.2 ±2.2 | 97.4 ±3.3 | 98.2 ±2.4 | **8.3** ±10.9 | 7.2 ±6.4 |
| | | $E$ | Energy | 91.2 ±8.6 | 89.5 ±13.0 | 80.9 ±25.9 | 44.7 ±32.7 | 22.3 ±16.7 |
| | | MSP | softmax | 92.0 ±7.4 | 89.5 ±12.2 | 84.2 ±19.5 | 40.7 ±26.4 | 23.1 ±16.7 |
| | Dist | TRUSTED | | **97.2** ±2.7 | **95.6** ±5.0 | **97.8** ±2.0 | **17.2** ±18.0 | 14.5 ±12.6 |
| | | $D_M$ | Pooled | 95.5 ±8.1 | 93.6 ±11.0 | 95.2 ±6.9 | 19.2 ±20.1 | **14.2** ±12.7 |
| | | $E$ | Energy | 88.2 ±9.4 | 86.4 ±13.3 | 77.4 ±27.0 | 57.0 ±34.3 | 30.5 ±22.0 |
| | | MSP | softmax | 88.6 ±7.9 | 84.9 ±15.4 | 78.2 ±22.7 | 56.7 ±26.5 | 32.1 ±21.7 |
| | Rob | TRUSTED | | **94.6** ±8.2 | **93.5** ±8.0 | **94.3** ±8.1 | 22.8 ±26.9 | 16.1 ±15.3 |
| | | $D_M$ | Pooled | 93.2 ±10.5 | 93.1 ±10.6 | 92.0 ±14.4 | **20.7** ±24.4 | **12.5** ±13.3 |
| | | $E$ | Energy | 91.4 ±9.2 | 89.9 ±10.6 | 83.3 ±19.2 | 38.6 ±28.9 | 22.3 ±17.2 |
| | | MSP | softmax | 91.9 ±6.4 | 90.3 ±7.7 | 85.9 ±14.3 | 38.1 ±25.3 | 23.2 ±15.9 |

Tab. 8: Average OOD detection performance (in %) per OUT-DS and per pretrained encoders.

## C.2 Analysis Per IN-Dataset

In Fig. 10, we conduct a dynamical analysis per IN-DS. We observe a variation of performance (*e.g.,* up to 10 AUROC points for sst2 for $D_M$) while probing across time.
**Takeaways:** Although our method achieves strong results a key dimension when deploying OOD detection methods is to carefully select the checkpoints when learning the classifier.

## C.3 Impact of the pretrained encoder

In Fig. 11, we report the results of the dynamical analysis and study the influence of the encoder choice on the detection performance.
**Takeaways** Although, TRUSTED achieves strong results on a large number of configurations. We observe different behaviours while considering different success criterion or different pretrained

| OUT DS | MODEL | TECH | Feature Type | AUROC | AUPR-IN | AUPR-OUT | FPR | Err |
|---|---|---|---|---|---|---|---|---|
| rte | BERT | TRUSTED | | **98.8** $_{\pm1.4}$ | **98.3** $_{\pm1.9}$ | 98.6 $_{\pm1.6}$ | **6.2** $_{\pm9.1}$ | **6.0** $_{\pm5.8}$ |
| | | $D_{\mathrm{M}}$ | Pooled | 98.6 $_{\pm1.5}$ | 97.8 $_{\pm3.0}$ | **98.8** $_{\pm1.2}$ | 6.4 $_{\pm6.9}$ | 6.0 $_{\pm4.3}$ |
| | | $E$ | Energy | 91.0 $_{\pm6.3}$ | 89.6 $_{\pm8.8}$ | 82.3 $_{\pm22.6}$ | 45.4 $_{\pm28.9}$ | 23.1 $_{\pm16.9}$ |
| | | MSP | softmax | 89.3 $_{\pm8.1}$ | 87.6 $_{\pm11.0}$ | 81.3 $_{\pm20.5}$ | 50.9 $_{\pm27.5}$ | 27.4 $_{\pm18.4}$ |
| | Dist | TRUSTED | | **99.0** $_{\pm0.7}$ | **98.4** $_{\pm1.6}$ | **98.3** $_{\pm3.1}$ | **4.4** $_{\pm3.9}$ | **4.9** $_{\pm2.5}$ |
| | | $D_{\mathrm{M}}$ | Pooled | 97.9 $_{\pm2.4}$ | 96.6 $_{\pm5.0}$ | 97.7 $_{\pm2.9}$ | 9.4 $_{\pm8.6}$ | 7.4 $_{\pm5.3}$ |
| | | $E$ | Energy | 90.2 $_{\pm7.9}$ | 89.8 $_{\pm9.9}$ | 79.6 $_{\pm24.4}$ | 47.5 $_{\pm33.5}$ | 23.4 $_{\pm19.3}$ |
| | | MSP | softmax | 89.8 $_{\pm7.3}$ | 88.9 $_{\pm9.8}$ | 79.6 $_{\pm22.0}$ | 50.6 $_{\pm27.5}$ | 26.1 $_{\pm18.0}$ |
| | Rob | TRUSTED | | **97.3** $_{\pm2.2}$ | **96.7** $_{\pm2.7}$ | **95.1** $_{\pm8.7}$ | **15.7** $_{\pm14.7}$ | **10.4** $_{\pm8.3}$ |
| | | $D_{\mathrm{M}}$ | Pooled | 92.0 $_{\pm11.3}$ | 88.9 $_{\pm17.1}$ | 92.5 $_{\pm11.3}$ | 19.1 $_{\pm19.4}$ | 11.7 $_{\pm11.5}$ |
| | | $E$ | Energy | 92.2 $_{\pm6.0}$ | 90.4 $_{\pm9.7}$ | 84.0 $_{\pm20.1}$ | 40.6 $_{\pm26.5}$ | 21.4 $_{\pm15.5}$ |
| | | MSP | softmax | 91.7 $_{\pm6.5}$ | 89.8 $_{\pm10.0}$ | 84.3 $_{\pm18.8}$ | 40.8 $_{\pm24.3}$ | 22.3 $_{\pm15.4}$ |
| sst2 | BERT | TRUSTED | | **98.5** $_{\pm1.8}$ | 98.1 $_{\pm2.8}$ | **95.7** $_{\pm6.2}$ | **9.1** $_{\pm15.6}$ | **8.1** $_{\pm12.1}$ |
| | | $D_{\mathrm{M}}$ | Pooled | 95.2 $_{\pm6.7}$ | **98.3** $_{\pm1.7}$ | 83.8 $_{\pm26.0}$ | 17.8 $_{\pm23.2}$ | 5.8 $_{\pm3.5}$ |
| | | $E$ | Energy | 89.3 $_{\pm12.9}$ | 95.7 $_{\pm3.8}$ | 72.4 $_{\pm39.8}$ | 38.3 $_{\pm38.2}$ | 11.7 $_{\pm7.9}$ |
| | | MSP | softmax | 90.1 $_{\pm10.4}$ | 95.2 $_{\pm4.2}$ | 73.0 $_{\pm38.5}$ | 37.9 $_{\pm32.7}$ | 12.5 $_{\pm5.3}$ |
| | Dist | TRUSTED | | **95.9** $_{\pm3.8}$ | 94.3 $_{\pm8.0}$ | **89.6** $_{\pm14.3}$ | **22.8** $_{\pm21.3}$ | 16.1 $_{\pm16.1}$ |
| | | $D_{\mathrm{M}}$ | Pooled | 92.3 $_{\pm8.3}$ | **96.2** $_{\pm4.3}$ | 77.9 $_{\pm29.5}$ | 28.5 $_{\pm25.8}$ | **10.2** $_{\pm7.9}$ |
| | | $E$ | Energy | 86.2 $_{\pm12.4}$ | 90.3 $_{\pm12.0}$ | 68.7 $_{\pm40.0}$ | 46.2 $_{\pm34.2}$ | 16.5 $_{\pm12.2}$ |
| | | MSP | softmax | 86.0 $_{\pm11.5}$ | 90.5 $_{\pm11.5}$ | 67.2 $_{\pm40.8}$ | 50.1 $_{\pm31.7}$ | 18.6 $_{\pm12.6}$ |
| | Rob | TRUSTED | | **95.4** $_{\pm3.6}$ | 95.0 $_{\pm6.0}$ | **87.9** $_{\pm19.5}$ | 27.4 $_{\pm23.3}$ | 16.1 $_{\pm14.6}$ |
| | | $D_{\mathrm{M}}$ | Pooled | 94.3 $_{\pm8.9}$ | **97.1** $_{\pm3.6}$ | 85.1 $_{\pm23.7}$ | **16.8** $_{\pm19.7}$ | **6.5** $_{\pm3.8}$ |
| | | $E$ | Energy | 89.4 $_{\pm10.1}$ | 93.6 $_{\pm7.3}$ | 72.9 $_{\pm37.1}$ | 42.1 $_{\pm31.7}$ | 15.9 $_{\pm11.2}$ |
| | | MSP | softmax | 89.6 $_{\pm9.4}$ | 93.9 $_{\pm5.9}$ | 73.0 $_{\pm36.6}$ | 41.5 $_{\pm27.4}$ | 15.5 $_{\pm7.5}$ |
| trec | BERT | TRUSTED | | **98.6** $_{\pm2.1}$ | **99.7** $_{\pm0.5}$ | **91.9** $_{\pm10.7}$ | **8.3** $_{\pm15.7}$ | **4.7** $_{\pm4.3}$ |
| | | $D_{\mathrm{M}}$ | Pooled | 92.4 $_{\pm9.2}$ | 97.9 $_{\pm3.6}$ | 69.2 $_{\pm25.1}$ | 31.0 $_{\pm28.5}$ | 9.6 $_{\pm6.4}$ |
| | | $E$ | Energy | 87.9 $_{\pm10.4}$ | 97.2 $_{\pm3.5}$ | 50.7 $_{\pm31.1}$ | 56.3 $_{\pm33.0}$ | 12.8 $_{\pm7.5}$ |
| | | MSP | softmax | 89.9 $_{\pm6.1}$ | 97.5 $_{\pm2.0}$ | 52.1 $_{\pm25.4}$ | 52.4 $_{\pm28.3}$ | 13.3 $_{\pm7.0}$ |
| | Dist | TRUSTED | | **96.8** $_{\pm4.0}$ | **99.2** $_{\pm1.1}$ | **83.3** $_{\pm14.6}$ | **20.0** $_{\pm27.0}$ | **8.2** $_{\pm5.9}$ |
| | | $D_{\mathrm{M}}$ | Pooled | 91.2 $_{\pm9.2}$ | 97.8 $_{\pm3.7}$ | 61.0 $_{\pm26.7}$ | 35.7 $_{\pm25.2}$ | 9.6 $_{\pm5.6}$ |
| | | $E$ | Energy | 88.0 $_{\pm9.6}$ | 97.2 $_{\pm3.1}$ | 46.1 $_{\pm29.8}$ | 57.6 $_{\pm38.0}$ | 12.9 $_{\pm8.3}$ |
| | | MSP | softmax | 88.6 $_{\pm8.2}$ | 97.2 $_{\pm2.8}$ | 45.2 $_{\pm26.0}$ | 55.1 $_{\pm30.0}$ | 13.3 $_{\pm7.8}$ |
| | Rob | TRUSTED | | **96.5** $_{\pm2.9}$ | **99.1** $_{\pm0.8}$ | **82.5** $_{\pm14.0}$ | **21.7** $_{\pm19.1}$ | **8.7** $_{\pm3.9}$ |
| | | $D_{\mathrm{M}}$ | Pooled | 92.9 $_{\pm9.4}$ | 97.7 $_{\pm4.0}$ | 80.8 $_{\pm20.5}$ | 23.7 $_{\pm22.2}$ | 8.5 $_{\pm4.6}$ |
| | | $E$ | Energy | 90.7 $_{\pm6.0}$ | 97.4 $_{\pm2.7}$ | 54.5 $_{\pm24.3}$ | 45.4 $_{\pm25.9}$ | 12.0 $_{\pm5.7}$ |
| | | MSP | softmax | 89.4 $_{\pm9.1}$ | 96.8 $_{\pm3.5}$ | 53.6 $_{\pm21.7}$ | 44.6 $_{\pm23.7}$ | 12.4 $_{\pm5.5}$ |
| wmt16 | BERT | TRUSTED | | 96.2 $_{\pm5.5}$ | 94.5 $_{\pm7.2}$ | **97.1** $_{\pm4.1}$ | 16.1 $_{\pm23.2}$ | 12.2 $_{\pm14.5}$ |
| | | $D_{\mathrm{M}}$ | Pooled | **96.6** $_{\pm4.3}$ | **94.6** $_{\pm7.6}$ | 96.9 $_{\pm3.4}$ | **13.3** $_{\pm14.5}$ | **9.8** $_{\pm9.2}$ |
| | | $E$ | Energy | 89.6 $_{\pm9.0}$ | 88.0 $_{\pm11.0}$ | 80.9 $_{\pm23.9}$ | 46.6 $_{\pm34.2}$ | 23.8 $_{\pm19.9}$ |
| | | MSP | softmax | 89.1 $_{\pm9.2}$ | 86.8 $_{\pm11.6}$ | 82.0 $_{\pm19.7}$ | 47.6 $_{\pm29.4}$ | 26.0 $_{\pm19.3}$ |
| | Dist | TRUSTED | | **97.3** $_{\pm2.7}$ | **95.3** $_{\pm4.5}$ | **97.6** $_{\pm2.7}$ | **11.8** $_{\pm12.4}$ | **9.8** $_{\pm7.4}$ |
| | | $D_{\mathrm{M}}$ | Pooled | 95.4 $_{\pm5.0}$ | 92.7 $_{\pm9.5}$ | 94.7 $_{\pm5.9}$ | 17.7 $_{\pm15.9}$ | 11.6 $_{\pm10.2}$ |
| | | $E$ | Energy | 89.6 $_{\pm9.7}$ | 87.8 $_{\pm12.2}$ | 80.5 $_{\pm23.2}$ | 45.0 $_{\pm33.3}$ | 22.9 $_{\pm19.4}$ |
| | | MSP | softmax | 89.1 $_{\pm8.6}$ | 86.9 $_{\pm10.9}$ | 79.5 $_{\pm21.5}$ | 49.1 $_{\pm28.1}$ | 25.8 $_{\pm18.4}$ |
| | Rob | TRUSTED | | **96.8** $_{\pm2.6}$ | **95.6** $_{\pm3.5}$ | **96.3** $_{\pm5.5}$ | **17.6** $_{\pm15.9}$ | 12.7 $_{\pm9.6}$ |
| | | $D_{\mathrm{M}}$ | Pooled | 90.4 $_{\pm13.5}$ | 88.4 $_{\pm17.2}$ | 91.0 $_{\pm13.9}$ | 21.4 $_{\pm21.0}$ | **12.5** $_{\pm12.0}$ |
| | | $E$ | Energy | 92.3 $_{\pm5.8}$ | 89.9 $_{\pm8.3}$ | 85.1 $_{\pm18.5}$ | 39.4 $_{\pm26.6}$ | 21.8 $_{\pm16.1}$ |
| | | MSP | softmax | 91.5 $_{\pm6.3}$ | 89.0 $_{\pm9.1}$ | 85.2 $_{\pm16.7}$ | 40.8 $_{\pm24.7}$ | 23.5 $_{\pm16.3}$ |

Tab. 9: Average OOD detection performance (in %) per `OUT-DS` and per pretrained encoders.

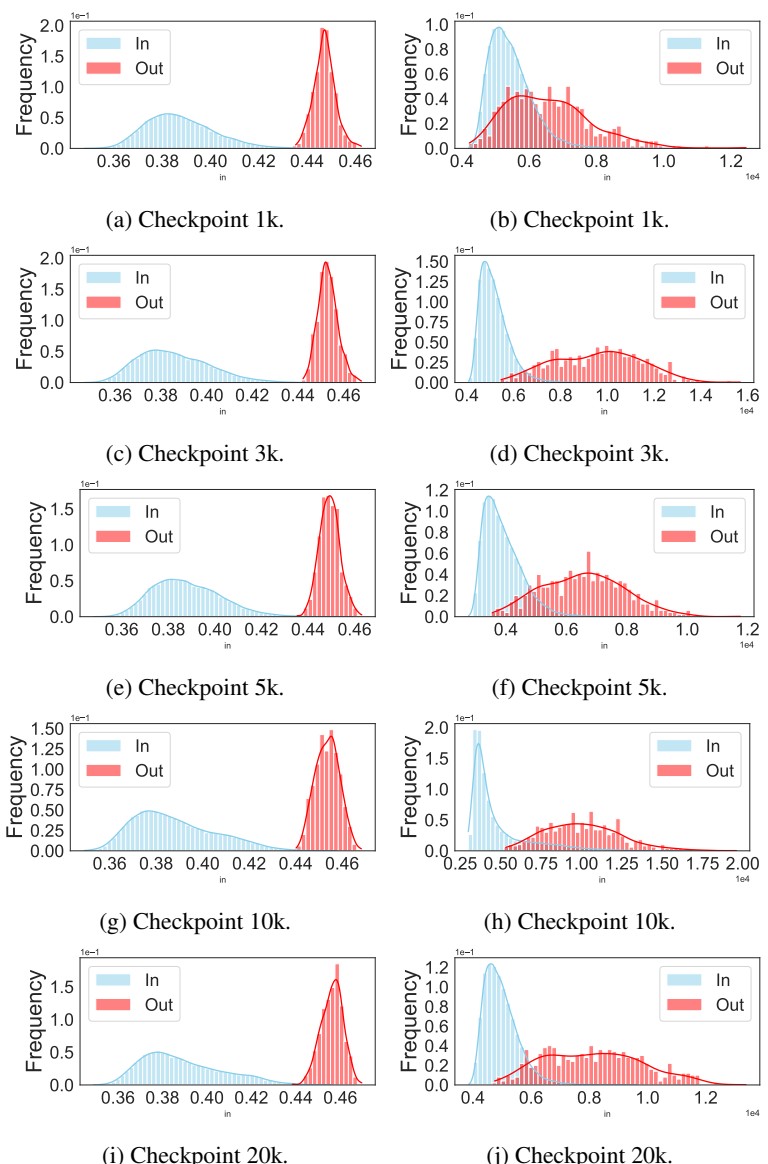

(a) Checkpoint 1k.  (b) Checkpoint 1k.

(c) Checkpoint 3k.  (d) Checkpoint 3k.

(e) Checkpoint 5k.  (f) Checkpoint 5k.

(g) Checkpoint 10k.  (h) Checkpoint 10k.

(i) Checkpoint 20k.  (j) Checkpoint 20k.

Fig. 9: OOD detection score histogram when the IN-DS is IMDB OUT-DS is TREC for various checkpoints. Left column corresponds to TRUSTED while the right column corresponds to Mahalanobis distance.

models. For examples, on BERT and DIS., TRUSTED is uniformly better across checkpoints when considering Fig. 11. For ROB., TRUSTED is not better on all metrics on last checkpoint (*e.g.,* 20k).

## C.4 All Combinations

For completeness of the paper, we report all the results of the dynamical analysis for all considered combinations in this section (see Fig. 12 Fig. 13 Fig. 14 Fig. 15). We believe this will allow the curious reader to gain more intuition and draw nuanced conclusions from our experiments.

## C.5 Futur works

Futur works include testing our method on different settings such as sequence generation, multimodal learning or automatic evaluation [21, 36, 53, 101, 43, 28, 15].

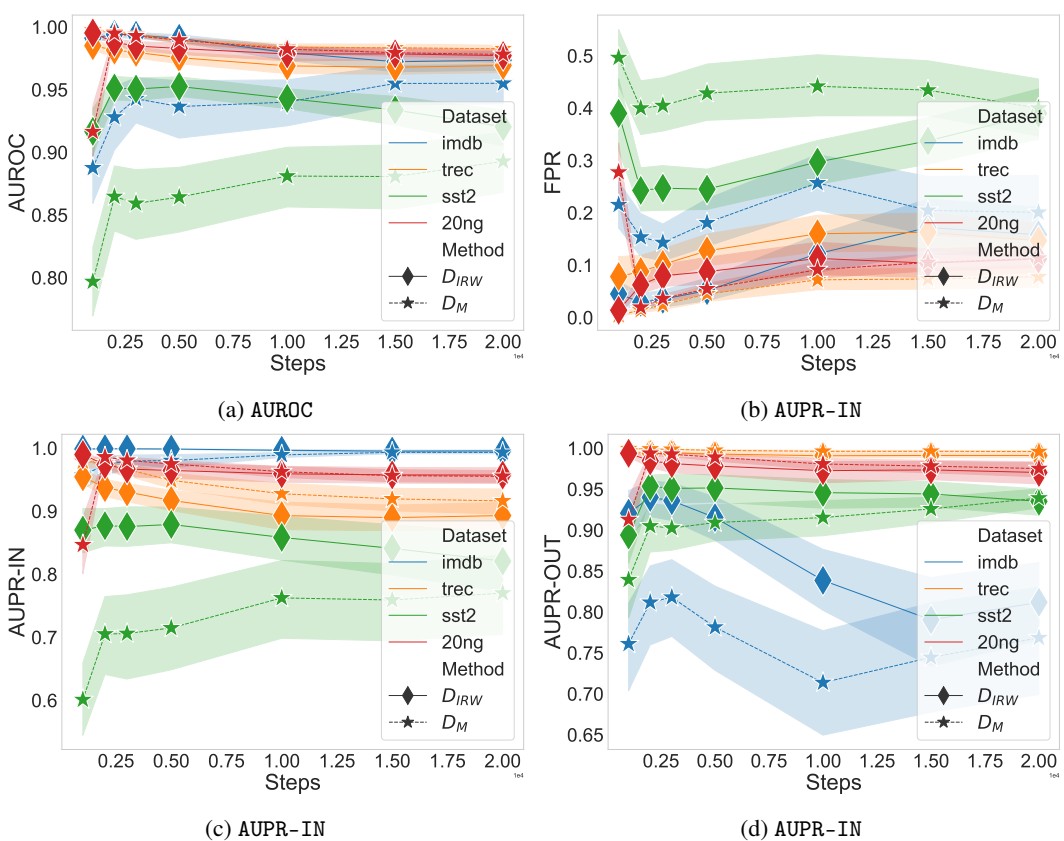

(a) `AUROC`

(b) `AUPR-IN`

(c) `AUPR-IN`

(d) `AUPR-IN`

Fig. 10: OOD performance across different checkpoints for the different `IN-DS`.

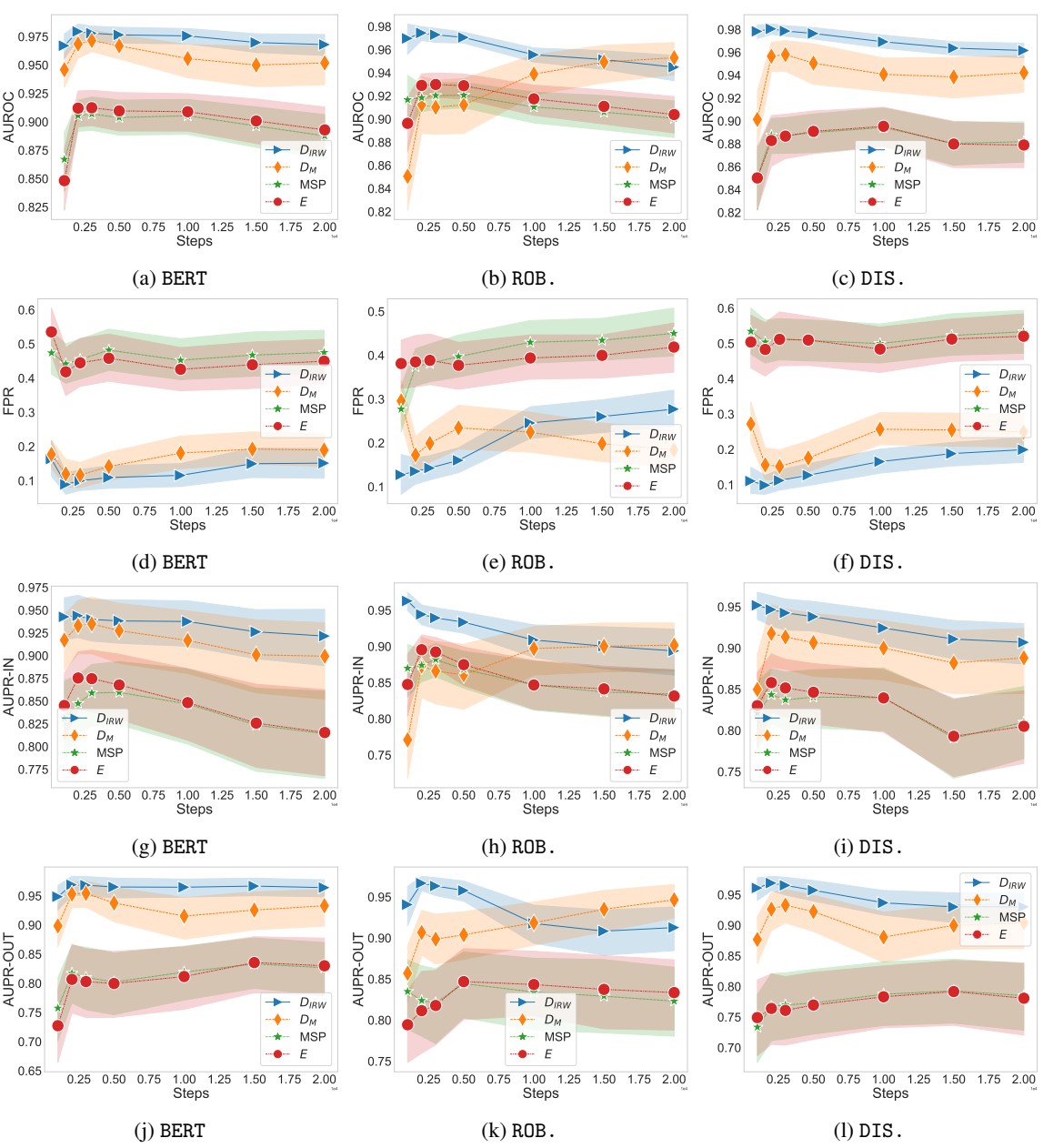

Fig. 11: OOD performance of the four considers methods across different checkpoints for the different pretrained models.

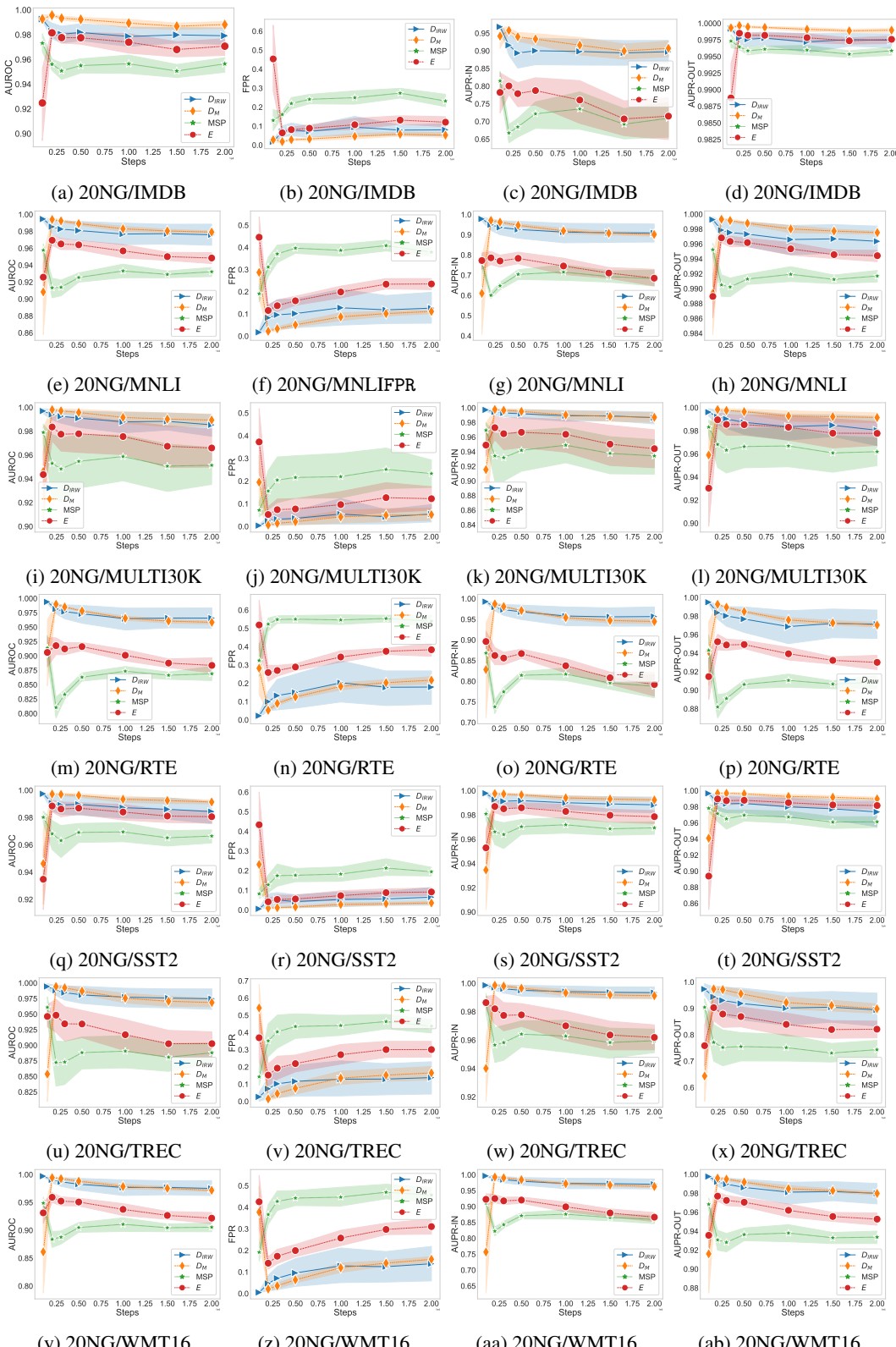

Fig. 12: OOD performance of the four considers methods across different `OUT-DS` for 20NG. Results are aggregated per pretrained models.

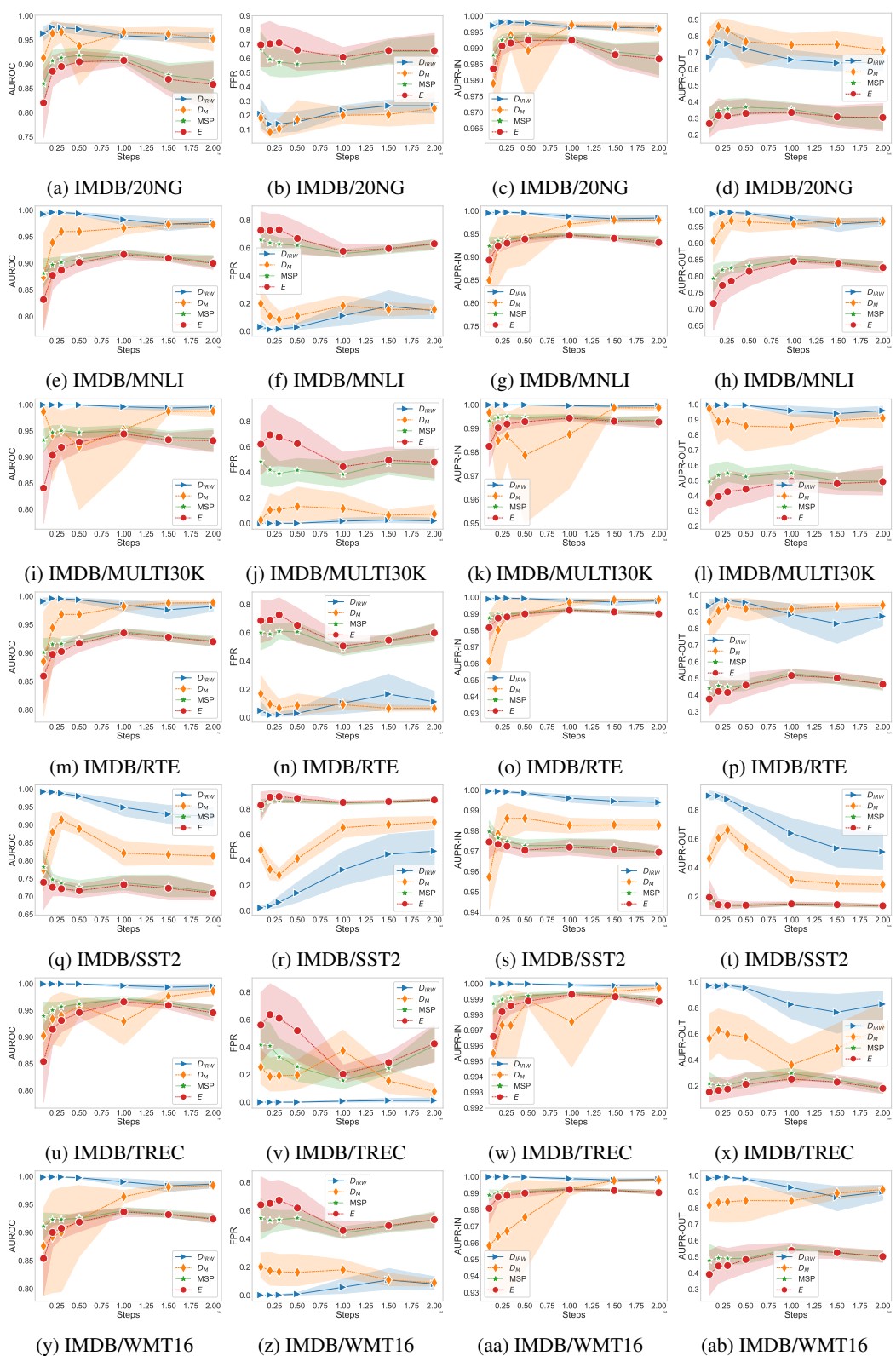

Fig. 13: OOD performance of the four considers methods across different `OUT-DS` for IMDB. Results are aggregated per pretrained models.

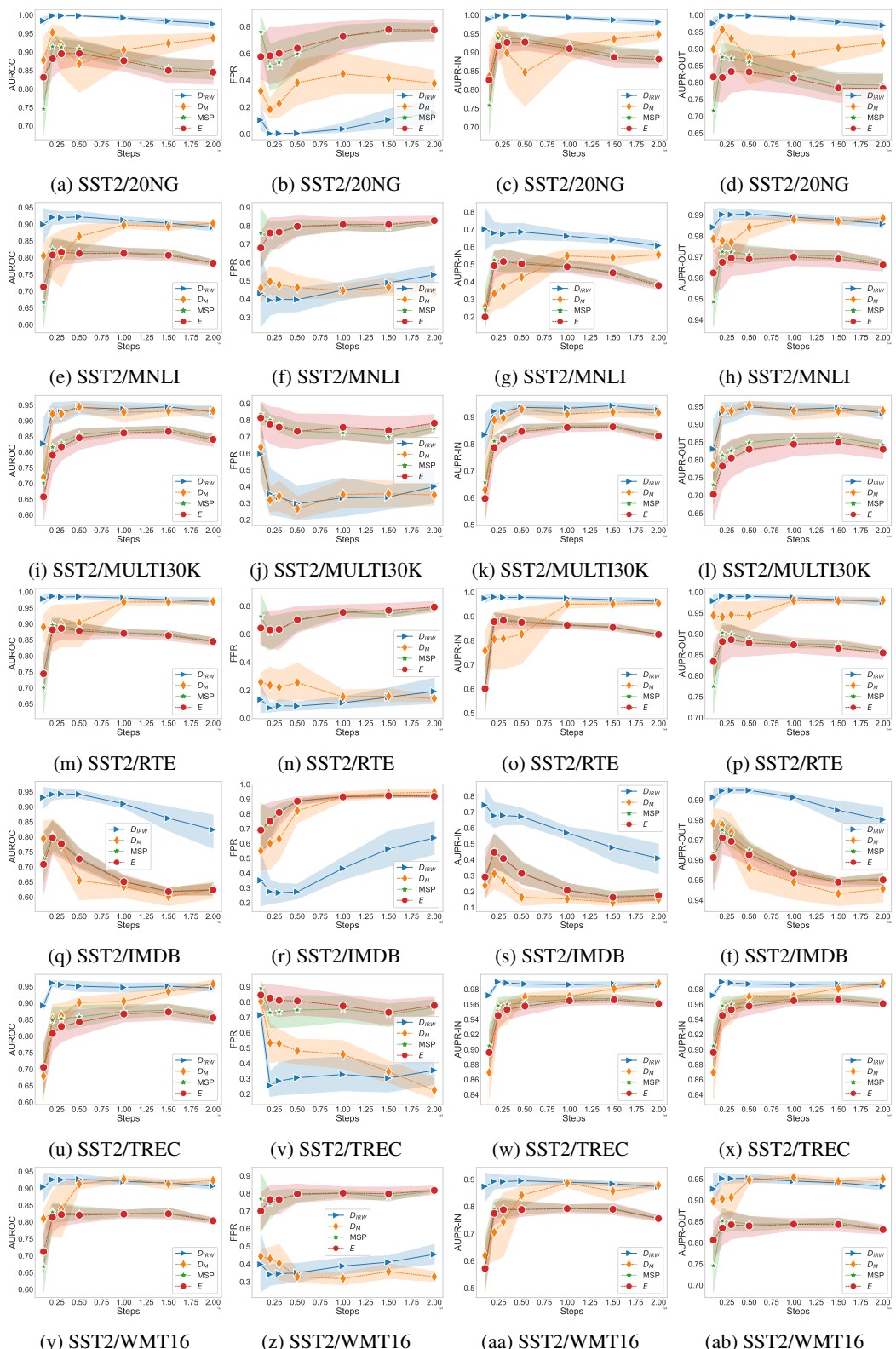

Fig. 14: OOD performance of the four considers methods across different `OUT-DS` for SST2. Results are aggregated per pretrained models.

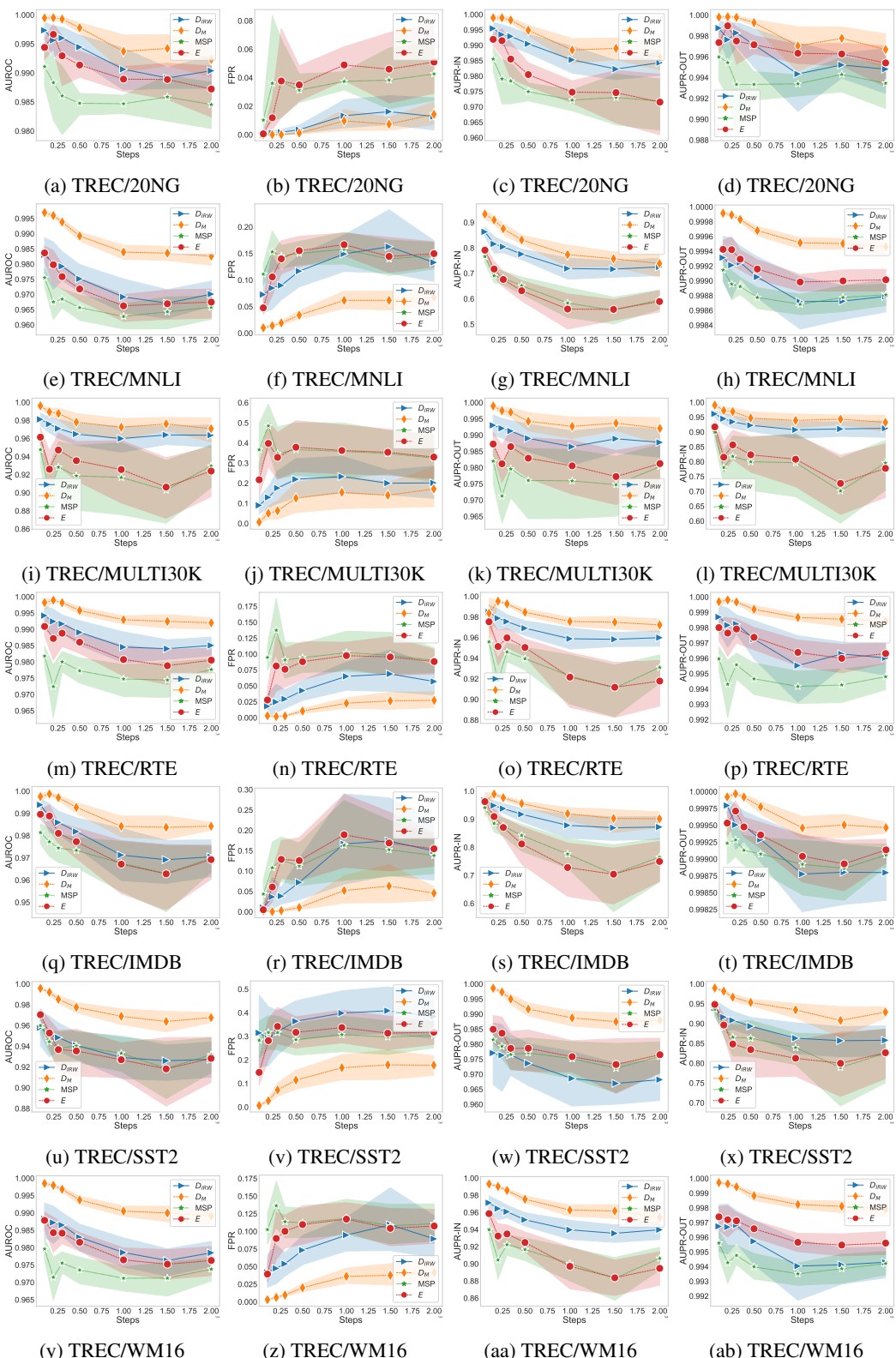

Fig. 15: OOD performance of the four considers methods across different `OUT-DS` for TREC. Results are aggregated per pretrained models.