# OpenReview forum: "Beyond Mahalanobis Distance for Textual OOD Detection"
_NeurIPS.cc/2022/Conference — NeurIPS 2022 Accept_

### Official Review · Reviewer_Ezx2 · 2022-07-11

**Rating:** 4
**Confidence:** 4
**Soundness:** 3 good
**Presentation:** 2 fair
**Contribution:** 2 fair

**Summary:**

This paper develops a new tool for out-of-distribution (OOD) detection in the NLP domain, which aims to complement the need of deformation techniques as that in computer vision. The key to detecting OOD relies on the computation of the OOD score that is realized by an integrated rank-weighted (IRW) depth.  Extensive experiments are conducted to show the superiority of the proposed method.

**Questions:**

1. As there are many different tools to solve the OOD problem, what is the exact motivation of borrowing the IRW depth to solve this problem?
2. The reviewer expects to see the connection of the proposed IRW depth and the deformation technique in CV.  Can the authors provide some theoretical results?


**Limitations:**

Yes

**Strengths And Weaknesses:**

Strengths:
1. This paper aims to solve an interesting topic.
2. Extensive experiments are conducted and show competitive performance for the proposed method.
3. This paper has delivered a thorough analysis of the related work.

Weakness:
1. The paper suffers from limited technical novelty.  For example,

1) The proposed IRW depth is borrowed for existing work.

2) The paper has claimed that one of the contributions is to aggregate all hidden layers of a neural network.  However, this kind of trick has been proposed in the literature.  The authors should cite the related work and highlight their main differences.

3) The reviewer doubts whether there is any theoretical guarantee on the proposed method.  In particular, the reason for choosing "the halfspace depth" in line 174 is unknown.

2. Some main technical implementations are not clear.  For example,

1) In line 149, the notation of \delta is not defined.  It makes it hard for the reviewer to know how to compute the similarity score and justify its correctness.  Moreover, \delta has appeared twice in the paper.  One is in lines 132-133, and the other is in line 149.  This makes the reviewer a little confused about understanding the meaning.

2) In line 147, the authors define F_{PM}(S) as the distribution of the mean-aggregation of the training distribution samples with the same predicted target as x.  However, it applies different aggregation strategies in Sec. 4.3, lines 246-255.

3) In line 159, “a pre-score aggregation function”: the definition is not shown.

4) In lines 182-183, u_k and n_{proj} are not defined.


3. Critical issues in the experiments:
a) Though the authors claim that the motivation for introducing IRW depth is to reduce the computational complexity, there are no such experiments to verify this issue.  The paper should address it through experimental support.
b)	In Table 2, it contains some contradictory conclusions and lacks sufficient experiment comparison.

i.	The paper claims the information from aggregating all layers in the neural network is more useful. However, for the D_M score in Table 2, the result of the last layer F_L is much better than of all layers F_{cat} and F_{PM}, which shows a contradiction result.  The authors have tried to explain the results in lines 269-270, i.e., the multivariate Gaussian assumption may not hold for the dataset.  It makes the reviewer confuse about the experimental comparison and makes unconvincing the experimental results because the aggregation of F_L is also under the Gaussian assumption and it beats F_{PM}.

ii.	In D_{IRW}, it only compares F_{cat} and F_{PM} with F_L and F_{L+1} and shows that  F_{cat} and F_{PM} beat F_L and F_{L+1}.  The comparison is not sufficient.  For example, the authors can consider other combinations, such as concatenating F_L and F_{L+1} or averaging F_L and F_{L+1}.

---

> ### Author Response · Authors · 2022-08-01
> **Reply to Ezx2**
>
> Let us thank Ezx2 for their careful reading of the manuscript. We are glad that Ezx2 acknowledges the strong performances of our detector, the extensive evaluation framework, our analysis and highlights the practical importance of designing OOD detectors for textual data.
>
> Below we address the different limitations:
> 1. Ezx2 claims that our work suffers from limited technical novelty:
>    * The IRW depth is both not known and not used in the deep learning community and more specifically in the NLP community. Although, we agree that IRW is borrowed from statistical literature, we argue that Mahanalobis distance applied to OOD detection in [44] has not been developed by the authors. We believe that finding, adapting and releasing new tools can benefit the NLP community.
>    * To the best of our knowledge, the idea of taking advantage of all hidden layers of transformers before computing the OOD score is actually new. First, in CV, OOD scores are computed for each layer and then aggregated into a single score (see Sastry and Gomes). In our work, we do not compute an anomaly score per layer but rather aggregate the hidden representations of samples across layers. We tested several approaches including [18] and [33] but went for the simplest solution. This is quite different and actually takes advantage of the transformer architecture, where all layers share the same dimension. In a nutshell, CV (i) computes an anomaly score for each layer and (ii) aggregates them, while we (i) aggregate hidden representations and (ii) compute a single anomaly score. This is extensively explained with relevant references in the Section 3.2 Layer aggregation choice.
>    * Ezx2 claims that there are no guarantees behind IRW depth (l174). In this paper, we use the IRW depth. It has been show in [R1] that the approximation of the halfspace depth suffers from the curse of dimensionality involving statistical rates of order O($(log(n)/n)^{1/(d-1)}$) (see Equation (12) in [R1]) when n is the sample size. In contrast, for the IRW depth, it has been shown in [2] that approximation of the IRW depth doesn’t suffer from the curse of dimensionality (see Corollary B.3 in [R2]). **The previous comments have been added to the manuscript.** Furthermore, as explained Line 174-178, the IRW depth inherits advantageous computational properties over the halfspace depth.
> 2. We have modify the manuscript to clarify technical implementations.
> 3. Let us answer the experimental concerns of Ezx2.
>    * **Computation time comparison between IRW and Mahalanobis depths** has been added to Appendix D. **These experiments show computational advantages of IRW over Mahalanobis.**
>     * (i) and (ii). Following the reviewer’s suggestions we include relevant additional comparison in Table 2. See results below
>
> |Method|AUC|FPR|
> | ---|---| ---|
> |DM FL | 93.8 ±9.8 | 89.2 ±20.1 |
> |DM FL+1 | 71.7 ±13.7|54.7±32.0|
> |DM F[L,L+1]|81.7±20.7| 60.7±20.0|
> |DM FL⊕L+1|83.6±10.6|61.9 ±39.3|
> |DM Fcat|90.4±11.5|84.0±22.1|
> |DM FPM|81.2±15.3|67.7±28.7|
> |DIRW FL|92.6 ±8.0| 88.5 ±17.7|
> |DIRW FL+1|82.4 ±14.0|77.2 ±24.0|
> |DIRW F[L,L+1]|95.5 ±10.0|91.2 ±15.0|
> |DIRW FL⊕L+1|95.9 ±10.0|91.0 ±20.0|
> |DIRW Fcat|96.1 ±4.9|91.8 ±14.0|
> |**TRUSTED FPM**|**97.0±4.0**|**93.2±11.5**|
>
> where F[L,L+1] consists in concatenating the 2 last layers, and FL⊕L+1 consist in averaging the two last layers. _**We have also improved and clarified our claim.**_
>
>
> Below we answer the different questions raised by Ezx2.
> 1. Motivation of using IRW:
>    * **IRW has appealing properties (See l 187).**  First, contrarily to Mahalanobis, it does not rely on a multivariate gaussian assumption, then it does not require the first two moments to be finite and to both compute and inverse $\Sigma$ in high dimension, which can be ill-conditioned in low data regimes. Low data regimes could be harmful in practical applications since OOD detectors consider per class statistics, thus to ensure that  $\Sigma$ is well conditioned we need to ensure that enough samples per class are available.
>    * **IRW is better suited for OOD detection**. TRUSTED achieves stronger results than a Mahalanobis-based detector. Additionally, our detector can be easily implemented and IRW possesses a simple geometric explanation (see Appendix).
>
> 2. Connection with “deformation technique in CV”. We work on textual data as stated by the title and the paper positioning. CV is outside of the scope of this paper.
>
> **To sum up, we think there could be a misunderstanding here: our work is an NLP contribution and we made no such claim as connections with CV. Nonetheless, we did our best to answer each concern of Ezx2, and kindly ask them to reconsider our paper as an NLP contribution. In the case Ezx2 is satisfied with our detailed response, we kindly ask them to upgrade their score.**
>
>
> **References:**
> [R1]Nagy. Uniform convergence rates for the approximated halfspace and projection depth.
>
> [R2]Staerman. Affine-Invariant Integrated Rank-Weighted Depth.

---

### Official Review · Reviewer_FF5o · 2022-07-11

**Rating:** 4
**Confidence:** 4
**Soundness:** 2 fair
**Presentation:** 2 fair
**Contribution:** 2 fair

**Summary:**

This paper proposes an OOD detector for Transformer models. Their motivation is that all hidden layers of transformer-based models contribute to OOD detection. They verify their method with extensive experiments and show the performance jump on AUROC.

**Questions:**

- Their method is based on the Mahalanobis score, which depends on the gaussian assumption. However, this is questionable; such an assumption still fits a real-world setting. Instead of the Mahalanobis score, I am wondering if the authors consider other feature-space scores for their method. Also, the computation burden or Mahalanobis also needs to be discussed.
- They use Integrated Rank-Weighted depth in their method, which has been used in some previous related works. This downgrade the novelty of the contribution in this paper.
- Woking on textual OOD problem is reasonable; however, since their methods rely on the transformer model, the natural question is, could it be applied to other domains with Transformer-based models, such as the vision domain?
- Their selected datasets have different domain characteristics. For example, IMDB is a movie review, while QA is another domain. I wonder if they consider some harder OOD settings during their experiments?
- It is interesting their observation that training longer the classifier hurts detection. Do the authors consider training those models from scratch without pretraining, for example, train 6-layer transformer vs. 12-layer transformer from scratch and show the power of their metric without pretrained weights.

**Limitations:**

The detector proposed in this paper is simple and effective. However, this method is still incremental and still has some room to be improved. The detector proposed in this paper is simple and effective. However, this method is still incremental and still has some room to be improved. For example, if this paper uses intermedia hidden layers, then if the pretrained model affects the use of hidden states needs to be discussed. Also, when it comes to the vision domain, could this method still work?

**Strengths And Weaknesses:**

- Their motivation that going deeper inside the network can improve OOD detection is somehow limited. Also, intermediate representation for detection tasks is not new, at least in the vision domain. This paper works on NLP domain but is still somehow incremental.
- Their observation that using all hidden layers is straightforward and reasonable, but simply taking an average of those latent representations is simply.

---

> ### Author Response · Authors · 2022-08-01
> **Reply to FF5o**
>
> Let us thank FF5o for their careful reading of the manuscript. We are glad that FF5o acknowledges the strong results of our detector.
>
> Below we address the concerns raised by FF5o.
>
> * FF5o argues our method is somewhat incremental, in particular by highlighting the fact that our aggregation procedure, used in our NLP context, is already known by the Computer Vision (CV) community. We would like to underline that aggregation techniques used in CV **are quite different from the one we are proposing here for several reasons:**
>    * First, in CV, anomaly scores are computed at each layer and then aggregated into a single anomaly score (see Sastry and Gomes). In our work, we do not compute an anomaly score per layer but rather aggregate the hidden representations of samples across layers. This is quite different and actually takes advantage of the transformer architecture, where all layers share the same dimension. In a nutshell, CV (i) computes an anomaly score at each layer and (ii) aggregates them, while we (i) aggregate hidden representations and (ii) compute a single anomaly score. _This is extensively explained with relevant references in the Section 3.2 Layer aggregation choice._
>    * Moreover, the way anomaly scores of each layer are aggregated in CV is somewhat intricate. For instance, Sastry rely on a complex ad-hoc heuristic. Another example is provided by Gomes and [44], who learn the best way to aggregate anomaly scores, which requires access to OOD data during training. _We would like to insist on the fact that our detector is unsupervised, meaning we **do not** need access to OOD data beforehand, making it more realistic for real-world scenarios._
>
> * To the best of our knowledge, the idea of taking advantage of all hidden layers of transformers for OOD detection is actually new. The fact that our method is simple, namely, compute an averaged representation, should not occult performances of our detector (which outperforms all previous works). **In fact, we think simplicity is the right road to building robust and understandable OOD techniques.** Large Language Models already suffer from a lack of explainability, and we humbly believe it is our mission as a community to build reliable tools to increase safety, in our case for OOD detection.
>
> We now turn to the questions of FF5o:
> 1. _Our method is not based on the use of Mahalanobis scores but on the IRW data depth,_ a tool we borrow from the statistics community. Our paragraph “Connection to the Mahalanobis-based score” is only here to show that the Mahalanobis-based detector can be grouped into the broader range of data depths methods in statistics.
> 2. The point of our paper is actually to say that this notion of data depth has not been considered by the NLP community (nor the deep learning or CV community, actually). Our work shows that IRW is better suited for our purpose as it achieves better performances. Thus, **we believe that our work is not incremental, both from a conceptual and pragmatic standpoint.**
> 3. Our method is indeed generic and could be tested on Images or other types of data. We believe it would be an entirely new work since NLP and CV both have their own particularities. _We chose to focus on NLP as it already implies a huge amount of experiments._
> 4. Our paper relies on the same benchmark as [36,82], which is the standard benchmark for OOD in NLP. It contains hard pairs (e.g., SST vs. IMDB) as well as easier pairs.
> 5. We would like to clarify our positioning here. Our goal is to have an impact on models used in the real world. Thus, we have chosen to focus on already trained networks, both small (e.g. DistillBERT) and large (e.g. BERT). We did not train models from scratch because it is known since [36] that models pre-trained on huge corpus and fine-tuned on specific tasks lend themselves better to OOD detection. This is in contrast with models that are trained from scratch on the downstream tasks solely as FF5o suggests, specifically when the dataset is small as is the case for TREC10, in which case a training from scratch struggles at learning complex textual patterns. Nonetheless, we think our extensive experiments cover different model sizes using more than 51k configurations (including for instance DistillBERT, BERT and ROBERTA).
>
>
>
> To sum up and answer FF5o limitations part, we think their review stands from a CV perspective and do not consider our work as an NLP contribution which has its own specificities. _**We kindly ask reviewer FF5o to reconsider its opinion on the paper by considering our contribution from this NLP perspective. Considering this positioning and our detailed responses, we kindly invite FF5o to revise their grade.**_
>
>
>
> _References:_
>
> Sastry et al. Detecting out-of-distribution examples with gram matrices. ICML 2021
>
> Gomes et al. An Information Geometry Approach to Out-of-Distribution Detection. ICLR2022.
>
> Fort et al. Exploring the limits of out-of-distribution detection. NeurIPS 2021.

---

### Official Review · Reviewer_gSwg · 2022-07-11

**Rating:** 7
**Confidence:** 4
**Soundness:** 3 good
**Presentation:** 3 good
**Contribution:** 3 good

**Summary:**

The authors propose a novel out-of-distribution (OOD) detector for text classification based on similarity scores from the training distribution. The goal is to identify OOD samples with an unsupervised and fast method by leveraging information from the hidden layers of the representation model.  The main contributions are: i) use of hidden representations to compute a distance score for OOd detection, ii) comprehensive comparison of models and hyper-parameters, and iii) open source for replicability. The study shows that the proposed method has competitive results compared to related work.

**Questions:**

Questions to the Authors

Please address the following questions during the rebuttal:

- Could you elaborate on the hyper-parameter selection and define the classifier architecture.
- During training, are the pre-trained models freeze?
- As an extra contribution, the authors can compute calibration and sharpness of the predicted probability outputs. Please speculate on the relation between well calibrated output probabilities and your method to identify OOD samples.


**Limitations:**

The authors have addressed limitations of the proposed approach.


**Strengths And Weaknesses:**

Strengths


- Clear description of background knowledge and related work needed to understand the proposed approach.

- Clear description of the proposed approach.

- The authors perform a comprehensive comparison with related work.



Weaknesses


- It is not clear how the parameter initialization  and selection of hyper-parameters could affect the method performance.

---

> ### Author Response · Authors · 2022-08-01
> **Reply to reviewer gSwg**
>
> Let us thank reviewer gSwg for their careful reading of the manuscript. _We are very glad they are acknowledging that our method shows competitive results compared to previous works and that they appreciate our efforts to provide a thorough description of the method and to explain how it compares to prior approaches._
>
> Below, we provide detailed answers to the questions raised by gSwg.
>
> * **Role of the hyperparameters.** Our choice of parameters is reported in Supplementary A3, along with training and validation curves (see Fig 7). Preliminary experiments on BERT have allowed setting the learning rate and dropout rate for all the models.
>
> * **Classifier architectures.** The classifiers are built on a pretrained encoder with a classification head (MLP). Both encoder weights and the classification head were fine-tuned during training.
>
> * TRUSTED does not require any additional hyper-parameters except during the depth computation. The parameters of the latter are set to their default values,  following what was done in [60].
> * Our method is based on the computation of the IRW depth of the aggregated information at each layer. Except for the fact that we compute this quantity conditioning  to the maximum softmax probability of an input sample, we do not rely on the softmax probabilities in the aggregation procedure. Thus, we expect calibration to have little impact on the performance of our method, in contrast with softmax-based methods such as MSP.
>
> **_We hope we have addressed the reviewer's questions and that they will be keen to consider raising their score accordingly._**

---

> > ### Comment · Reviewer_gSwg · 2022-08-08
> > **Rebuttal**
> >
> > Thank you, the authors have addressed my questions during rebuttal.

---

### Official Review · Reviewer_o5Ud · 2022-07-12

**Rating:** 6
**Confidence:** 3
**Soundness:** 3 good
**Presentation:** 3 good
**Contribution:** 2 fair

**Summary:**

This work proposes TRUSTED: an out-of-distribution detection approach that uses integrated rank weighted (IRW) depth to assess whether an example falls within a set of in-distribution samples in the embedding space of a transformer model. The average embeddings across all transformer layers are used for optimal performance. TRUSTED is compared against 1) integrated rank weighted depth using final embedding layer, 2) Mahalanobis distance instead of IRW and 3) maximum softmax probability on a text OOD benchmark and shows superior performance.

**Questions:**

See Weaknesses.

**Limitations:**

Limitations are sufficiently addressed

**Strengths And Weaknesses:**

Strengths:
1. Approach and experiment settings are clearly explained.
2. Code is provided for reproducibility.
3. The detector can be applied to new transformer models without any training, as opposed to data-driven methods.

Weaknesses:
1. Experiments didn't show comparison against (any) existing works.
- [36]
- [82]
- Arora, Udit, William Huang, and He He. "Types of Out-of-Distribution Texts and How to Detect Them." EMNLP 2021.

2. Approach is fairly simplistic. Novelty is limited. More suitable as a "strong baseline" type of approach.
- There's even very limited contributions to benchmarking of text OOD detection given that approach is simple. At the current state, a sound & large OOD benchmark with diverse text, different types of OOD, with text from multiple domains may be the easiest way to boost the contributions of this work.

3. Questions:
- Table 2 uncertainties are large even after averaging over 1440 configurations. Why is that?

=========================

With comparisons to existing works, the authors have adequately addressed my strong concerns on the experiments. I have updated the score.

---

> ### Author Response · Authors · 2022-08-01
> **Reply to reviewer o5Ud**
>
> We thank reviewer o5Ud for their careful reading of the manuscript. We are glad they are acknowledging that our method _is well-suited for OOD detection: (i) when considering already trained models where retraining is not affordable, and (ii) when no OOD data is available (our method does not request OOD data)._ In this setting, we are glad that, in their summary, reviewer o5Ud underlines that our proposed method outperforms previously developed detection methods for textual data.
>
> In the following, we provide detailed answers and address each issue reviewer o5Ud raised.
> * Reviewer o5Ud asks for comparison against detector methods of [36], [82], and Arora’s paper. **In fact, we have already included the aforementioned methods in our benchmark.** More precisely:
>
>     * In [36], the authors study the benefits of using pretrained models on the performance of OOD detectors. They rely on the Maximum Softmax Prediction _(MSP) detector that is one of the baselines to which we compare  in our paper_ (see Tab 2 and Tab 3).
>
>     * The main contribution of [82] is a modification of the fine-tuning objective using contrastive learning (see Algo 2 in [82]). They study the effect of the new training on the performance of three OOD detectors, namely MSP, Energy, and Mahalanobis. _These detectors are used as baselines in our paper._ Indeed, we did not consider the detector based on the cosine distance (which is also used in [82]) as it has a lower performance than Mahalanobis-based method.
>
>     * In their paper, Arora et al. consider MSP and Perplexity. We do not use Perplexity as it corresponds to a different setting than ours. Indeed, we consider trained classifiers and try to detect OOD inputs, while _Perplexity is used for language models_ and compute the likelihood of the next word in a masked sentence.
>
>
> * There are two subpoints here.
>
>      * We disagree with reviewer o5Ud and think our method is, in fact, novel and brings a new tool to the NLP and  deep learning community. We introduced  the Integrated Rank-Weighted (IRW) depth which falls into the more general concept of data depths. The only data depth which was used for OOD detection in NLP is the Mahalanobis score. In fact, a contribution of our paper is to reveal that this score is actually a data depth. Based on this remark, we suggest using the Integrated Rank-Weight depth, which turns out to be much more efficient for our purpose. The IRW depth has never been used for textual data nor studied for OOD detection using deep neural networks. _So clearly, this contribution is novel._
>
>      * Our paper relies on the same benchmark as [36,82]. However, to the best of our knowledge, our benchmark is the largest experimental study conducted on OOD detection for textual data. Indeed, in our experiments, we benchmark OOD detectors over 51K models. Specifically, we consider 3 different encoder pre-trained models (namely BERT, DistillBERT, Roberta), which are trained on 4 different in distribution datasets (20ng, SST2, TREK, IMBD) and consider 8 out-of-distribution datasets (20ng, SST2, TREK, IMBD, Multi30K, MNLI, RTE, WMT16). For each model, we consider 5/7 different checkpoints and 3 seeds. As a comparison, related work such as [48, 68, 39, 53, 59] considers at most 2 models on a single checkpoint.
> Besides, our evaluation in Section 6 investigates the effect of checkpoints and choice of random seed and reveals interesting and previously unreported factors of variability across performances. We think this variability with respect to checkpoints and seeds is actually an important contribution to our work because it calls for new evaluation practices when evaluating OOD detectors.
>
> * As mentioned in the previous paragraph, _the variability of OOD detector's performance is actually one of the takeaways of our work._
>
>
> We humbly believe the concerns raised by o5Ud are somewhat not correlated with their score, especially considering that they recognize the superiority of our method over previous works. **_We hope very much that our above-detailed explanations address their questions and that they will be open  to consider raising their score accordingly._**

---

> > ### Comment · Reviewer_o5Ud · 2022-08-02
> > **Reply to author rebuttal on benchmark comparisons**
> >
> > For comparison to existing works, please provide comparison that
> > 1) On the same dataset & split the respective papers used
> > 2) Directly to the numbers in the tables reported in the respective papers (no reimplementation, or somehow prove that the reimplementation is the same or better)
> > 3) Use the proposed approach in the respective papers and cite the paper for the name of the approach in the table
> >
> > I have come to an understanding that
> > 1) Such a comparison against [36] is difficult because [36] provided only bar plots and didn't provide numbers for a straight comparison.
> > 2) Such a comparison against [85] may arguably be apples to oranges because [85] finetunes the transformer where as the proposed work did not finetune the transformer. (Apologize for the typo in the review, [82]->[85])
> > Note: I do not buy the argument that MSP, energy or Mahalanobis counts as a proper comparison to [85] since they are baseline approaches used in [85].
> > 3) Same for Arora et al, since they use a perplexity baseline, where as perplexity may be inaccessible in the textual classification problems that the authors are working on.
> > 4) Plus, the BERT models trained for the respective datasets may be non-standard and difficult to reproduce.
> >
> > As a result, no such straight head-to-head comparison can be performed, and the comparisons defaults to showing that proposed approach wins over a set of hard baselines.
> >
> > However I think that
> > 1) It would be very useful to apply the proposed approach *directly to pre-existing benchmarks* and show improvements, e.g. one from Arora et al.'s paper. It's the easiest way to provide a direct reference point to existing literature.
> > 2) Either a head-to-head comparison is needed ignoring some assumptions, or the proposed method is SOTA in only restricted conditions and the contribution would be less.
> > 3) If no such head-to-head comparisons are presented, the inherent assumptions should be made clear, for example "no finetuning" or "no perplexity" so comparisons to other contemporary methods may not be applicable. For example, emphasize OOD on pretrained models in the intro/abstract.

---

> > > ### Author Response · Authors · 2022-08-03
> > > **Reply to reviewer o5Ud**
> > >
> > > Let us thank reviewer o5Ud for their detailed answer to our response. We are glad they are acknowledging that a direct comparison against [36] and [85] is either not realistic in our setting, or outside of the scope of the paper.
> > >
> > >
> > > In the following, we would like to (i) _clarify our positioning as suggested by reviewer o5Ud_ and (ii) _provide them with additional experiments we run to take into account its suggestions._
> > >
> > >
> > > **Clarification of our positioning.** We are addressing the OOD detection problem on pre-trained and fine-tuned classifiers. _We believe this setting is particularly well suited for real-world applications where practitioners_ usually have already access to a trained model and cannot afford to do a retraining from scratch. In order to avoid confusion, we have added a clarification both in the abstract and in the introduction of the updated manuscript.
> > >
> > > **Extending our experimental setting to LM.** Although we think extending the use of TRUSTED to Language Models (such as in Arora et al.) is somewhat outside the scope of our work, _we did our best to add relevant experiments to address reviewer o5Ud._ More precisely, we implemented the code provided by Arora et al. to allow for a fair comparison. The results we obtained are displayed in the following Table:
> > >
> > > |ID| OOD| ||AUROC|| |Accuracy||
> > > |-|-|-|-|-|-|-|-|-|
> > >  |||     **_APPROVED_**| PPL | MSP|  Oracle|       OOD| IN-D|
> > > |SST-2| IMDB| **97.9**   |  96.8 |66.1| 100.0   |     91.8|  92.9|
> > > ||      Yelp |99.0 |    99.0| 57.3 |99.8 |        94.3||
> > > |IMDB |SST-2 |**98.9**  |   96.8 |83.1| 100.0   |     90.1|  95.7|
> > > ||Yelp | **88.9** |   76.5 |67.8 |100.0   |     96.2||
> > > |Yelp |SST-2 |**99.7** |    98.8| 85.8| 99.8       |  88.9 | 98.1|
> > >   ||   IMDB | **90.8**  |   86.7 |62.3| 100.0    |    93.1||
> > > |SNLI |RTE  | **97.9**   |  95.1| 78.7 |99.8    |     67.9|  90.2|
> > >  ||    MNLI | **98.6**     |96.4| 75.6| 99.7  |       80.1||
> > > |RTE  |SNLI | **87.4**   |  82.3 |47.1 |99.7      |   81.8 | 76.8| |
> > >   |  | MNLI  |**89.2**   |  85.3| 56.7| 97.0 |   77.1| |
> > > |MNLI |SNLI|  **89.6** |    76.6 |58.2| 99.7    |     81.3|  85.4|
> > >    ||  RTE|   **87.9**   |  68.2 |77.7 |96.7       |  77.1        ||
> > > |TOTAL  | |   **_93.8_**   |  88.1 |68.3| 99.3 ||||
> > >
> > >
> > > We would like to stress that this _Oracle detector is trained with access to OOD data and thus cannot be compared with our TRUSTED detector_ which does not use an OOD example. We nonetheless report the results of Oracle for completeness.
> > >
> > > **Clearly, these additional results support the superiority of our TRUSTED detector, even in the case of LM.**
> > >
> > > **We have added this experience (see Section B of the appendix) as an extension of our method. Upon acceptance we will carefully describing the corresponding new LM setting, the baselines and the citations of relevant works in the main paper.**
> > >
> > >
> > > **_We hope we have addressed the reviewer's questions on comparing with relevant related works and that they will be keen to consider raising their score accordingly._**

---

### Meta-Review · Area_Chair_YGXu · 2022-08-27

**Recommendation:** Accept
**Confidence:** Less certain

**Metareview:**

The paper proposes a out-of-distribution detection approach using integrated rank weighted (IRW). Its main novel feature is leveraging the information from all layers of the model for this task. The detector can be applied to new transformer models without any training, as opposed to data-driven methods. The method is assessed in a comprehensive evaluation and code is provided for reproducibility. One of the limitations, however, is the difficulty to compare the proposed method with related work.  Presentation, especially of the technical content, should be improved in the final version.

The AC disagrees with the authors' complaint about the biasedness of some reviews. Indeed two reviewers had critical remarks on several aspects of the paper, yet this criticism appears to be fair and driven by the scientific discourse. The answers provided in the rebuttal have clarified most of the reviewers' concerns.

**Award:**

No

---

### Decision · Program_Chairs · 2022-09-14

Accept